# Environmental motion presented ahead of self-motion modulates heading direction estimation

Liana Nafisa Saftari[1,2☯], Jongmin Moon[1☯¤], Oh-Sang Kwon[1*]

1 Department of Biomedical Engineering, Ulsan National Institute of Science and Technology, Ulsan, South Korea, 2 Biomedical Engineering Study Program, School of Electrical Engineering, Telkom University Main Campus, Bandung, Indonesia

☯ These authors contributed equally to this work.
¤ Current address: Center for Perceptual Systems, The University of Texas at Austin, Austin, Texas, United States of America
* oskwon@unist.ac.kr

## Abstract

The ability of a moving observer to accurately perceive their heading direction is essential for effective locomotion and balance control. While previous studies have shown that observers integrate visual and vestibular signals collected during movement, it remains unclear whether and how observers use visual signals collected before their movement to perceive heading direction. Here we investigate the effect of environmental motion that occurred ahead of self-motion on the perception of self-motion. Human observers sat on a motion platform, viewed visual motion stimuli, and then reported their perceived heading after the platform moved. The results reveal that environmental motion presented before the observers' movement significantly modulates their heading perception. We account for this effect using a normative computational model that takes into account the causal relationship between visual signals generated before and during the observers' movement. Overall, our study highlights the crucial role of environmental motion presented before self-motion in heading perception, broadening the current perspective on the computational mechanisms behind heading estimation.

## Author summary

Perceiving our own movement, such as walking down the street or trying to keep our balance, requires the brain to interpret noisy and ambiguous signals from our senses. This becomes especially challenging when the environment is also in motion, because the movement we see might result from either our own movement or something in the surroundings. In this study, we asked whether the brain could use visual motion signals gathered before we start moving to help resolve this ambiguity. Using a novel experimental paradigm, we found that motion in

**Data availability statement:** The data and code used in this study are publicly available on OSF (DOI: 10.17605/OSF.IO/RMFE6).

**Funding:** This work was supported by the National Research Foundation of Korea (NRF-2023R1A2C1007917 to O-SK). The funders had no role in study design, data collection and analysis, decision to publish, or preparation of the manuscript.

**Competing interests:** The authors have declared that no competing interests exist.

the environment, presented just before the self-motion, can change the way we perceive the direction of our movement. To understand why this happens, we developed a computational model grounded in principles of causal inference. The model captures how an ideal observer would estimate self-motion from sensory signals collected over time, given their belief about whether motion in the environment has remained constant. Together, our results indicate that the brain does not rely only on what's happening during movement but also incorporates visual temporal context to make optimal estimates of self-motion.

## Introduction

Heading perception is crucial for spatial navigation and balance control. Accurate heading perception becomes especially challenging when the surrounding environment is also in motion, as visual signals collected by the observer could originate from their own motion or from the motion in the environment [1–3]. Consider an observer who is standing up from a bench while watching a nearby bus on the road. If the bus appears to be moving down and to the right on her retina while she is getting up, it could be because the observer is moving vertically upward while the bus is moving to the right, or alternatively, the bus is stationary, but the observer misaligns her movement, tilting slightly to the left.

To resolve this ambiguity, the observer can use extra-retinal information obtained by the vestibular system [1–7]. If the vestibular signal confirms the vertically upward movement of the observer, the rightward motion of the bus on the retina is likely due to the bus moving in the world. Indeed, numerous previous studies have shown that observers use vestibular signal to separate environmental motion from self-motion [8–15]. Another effective way to resolve this ambiguity is by relying on temporal context. In the real world, buses do not suddenly appear on the road. Instead, the bus was likely already moving before the observer began to stand. Therefore, the observer can accurately perceive her heading direction by subtracting the previously presented motion of the bus from the visual motion signal collected during her rise.

As illustrated by this example, moving observers in a non-stationary environment need to appropriately use visual signals they collected before they start moving to accurately perceive their current heading direction (Fig 1A). While many studies have explored how observers integrate visual and vestibular signals collected during movement [16–23], much less is known about whether and how observers use visual signals collected before their movement to estimate the heading direction, if the surrounding environment is already in motion.

To address this question, we designed a psychophysical experiment that emulates the scenario described above (Fig 1B–1D). Observers, seated on a motion platform, were presented with visual motion stimuli, and then, as the visual motion continued, they were moved by the platform. The critical manipulation was that the visual motion stimuli presented before self-motion varied systematically across conditions, while the visual motion stimuli presented during self-motion were the same across

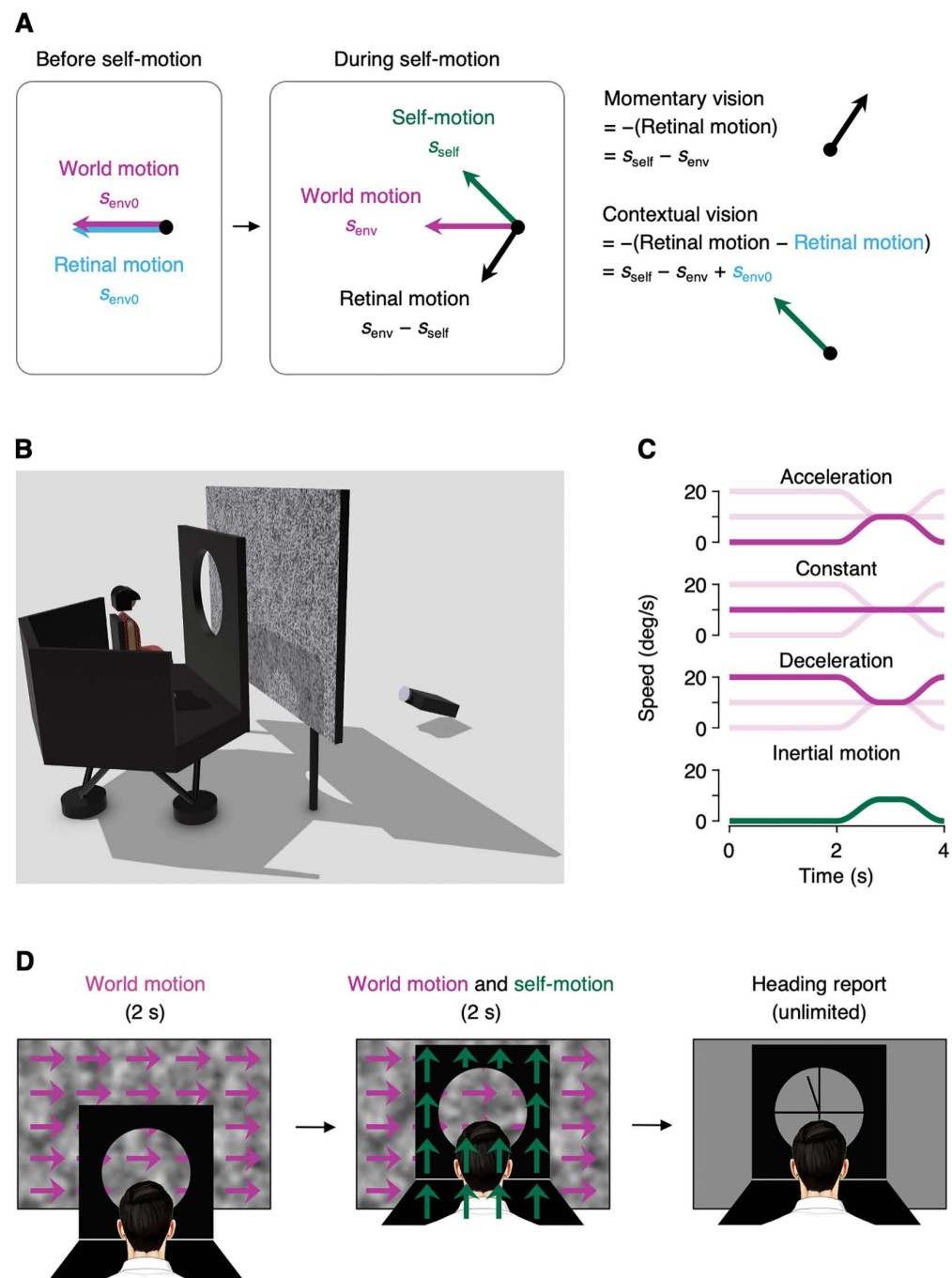

**Fig 1. Experimental paradigm.** (A) Conceptual framework. Before an observer begins to move, retinal motion is solely due to motion in the world (left). As the observer starts moving while the world motion continues, retinal motion reflects the vector subtraction of world motion and self-motion (center). If the observer interprets retinal motion at face value, their heading estimate will be opposite to the retinal motion (i.e., momentary vision). If the observer has a good reason to believe that world motion has remained constant before and during self-motion, they can estimate their heading by subtracting retinal motion before self-motion from retinal motion during self-motion (i.e., contextual vision). (B) Experimental setup. Observers sat on a motion platform and viewed visual motion stimuli on a rear-projection screen through a circular aperture. The screen was positioned outside the motion platform, while the aperture was mounted on and moved together with the motion platform. (C) Example speed profiles for visual motion stimuli (magenta) across three conditions (i.e., Acceleration, Constant and Deceleration) and for inertial motion stimuli (green). For comparison, the speed profiles of the other two visual motion conditions are shown in light magenta. (D) Experimental task. Observers were shown leftward or rightward visual motion (indicated

by magenta arrows, not shown in the experiment) for two seconds. After the two seconds, as visual motion continued, observers were passively moved along one of ten directions in the frontal plane ranging from −45° to 45° relative to vertically upward (indicated by green arrows, also not shown in the experiment). Following the synchronized offset of both visual and inertial motion stimuli, observers reported their perceived heading direction by adjusting a probe on the screen using a computer mouse.

conditions. We found that this manipulation caused a systematic difference in heading perception, highlighting the crucial role of environmental motion that occurred before self-motion. Using an optimal causal inference model, we propose that observers inferred the causal relationship between visual signals collected before and during self-motion, performing necessary computations given the inferred causal relationship, which led to the significant effect of environmental motion presented before self-motion on heading perception.

## Results

### Environmental motion presented ahead of self-motion modulates heading direction estimation

Human observers sat on a motion platform and viewed visual motion stimuli projected on a screen positioned outside the platform through a circular aperture attached to the platform (Fig 1B). On each trial, observers passively moved for two seconds in one of ten directions in the frontal plane, ranging from −45° to 45° relative to vertically upward, and reported their perceived heading direction (Fig 1D). Visual motion stimuli, moving either leftward or rightward, were presented through the aperture, beginning two seconds before the onset of the inertial motion and continuing until the end of the inertial motion (Fig 1D). We introduced three visual motion conditions, each differing in the velocity of visual motion presented before the inertial motion (Fig 1C). In Acceleration condition, the visual motion velocity was zero before the inertial motion. In Constant condition, the visual motion velocity remained constant before and during the inertial motion. In Deceleration condition, the visual motion velocity before the inertial motion was twice the visual motion velocity during the inertial motion. Importantly, the velocity of visual motion during the inertial motion was held constant across all conditions (either 0°/s, ±5°/s or ±10°/s). This design enabled us to isolate the effect of visual motion presented before self-motion while controlling for the effect of visual motion presented during self-motion.

Observers' heading estimation behavior exhibited three characteristic features (Fig 2; see also Fig B in S1 Supporting information for the group average). First, heading estimates were systematically biased in the direction opposite to the visual motion stimuli. When visual motion stimuli moved leftward, heading estimates were biased clockwise, and when visual motion stimuli moved rightward, heading estimates were biased counterclockwise. Notably, observers did not fully compensate for the motion in the environment, leading to a biased heading estimate even when the environmental motion remained constant before and during self-motion. Second, for each visual motion condition, the strength of heading biases depended on the speed of visual motion stimuli, such that heading biases were more pronounced with faster visual motion stimuli. Third, the strength of heading biases depended on visual motion stimuli presented ahead of self-motion. Specifically, heading biases were more pronounced in Constant condition than in Deceleration condition, and even more so in Acceleration condition.

We calculated the average angular difference between the true and perceived heading direction, with errors realigned such that positive deviations are in the direction of the visual motion stimuli (Fig 3B, dark blue). For example, if the visual motion stimuli moved rightward, a negative heading bias indicates that the heading estimates are more leftward than the true heading direction in the frontal plane. We found that the heading estimates are robustly biased away from the direction of visual motion stimuli, with a strong influence of the speed of visual motion ($F_{2,26} = 28.003$, $P < 0.001$). More importantly, we also found that the heading bias varies systematically depending on the visual motion condition, with a significant main effect of visual motion condition ($F_{2,26} = 46.373$, $P < 0.001$) and a significant interaction between visual motion condition and the speed of visual motion ($F_{4,52} = 28.134$, $P < 0.001$). Because the visual motion stimuli were exactly

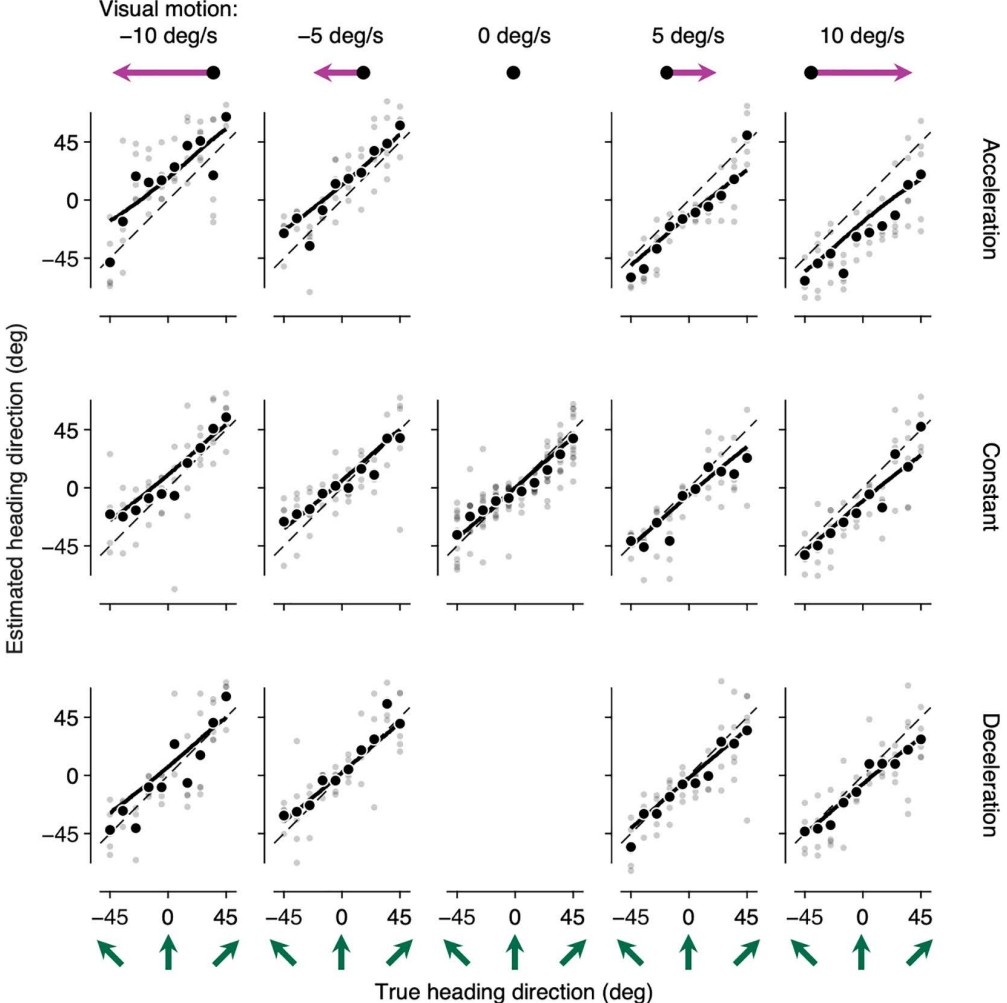

**Fig 2. Overview of the heading estimation behavior.** A representative human observer's heading estimates for every trial (small semi-transparent dots) and their averages for each heading direction (large filled circles) for each heading direction are shown for five visual motion velocities (plotted in each column) and three visual motion conditions (plotted in each row) along with the prediction of the Contextual Causal Inference model (solid lines). Heading estimates deviate on average from the unity line (diagonal dashed line) depending on visual motion velocities and visual motion conditions.

the same during self-motion across three visual motion conditions, the observed effect of visual motion condition indicates that the visual motion presented before self-motion modulated the heading perception.

To gain insights into why the observers' heading estimates are biased, we considered three sources of information that observers may utilize (Fig 3A). First, observers have vestibular information. Relying solely on the vestibular information would result in heading estimates centered around the true self-motion direction, $s_{self}$, without any heading bias due to visual stimuli, regardless of visual motion conditions (Fig 3B, purple). Second, observers may rely on retinal motion during self-motion, assuming that the environment is stationary while they are moving (i.e., momentary vision). Relying solely on the momentary vision would result in heading estimates centered around self-motion subtracted by environmental motion, $s_{self} - s_{env}$, leading to a strong bias away from the visual motion stimuli. (Fig 3B, yellow). Finally, observers may subtract the environmental motion before self-motion from the retinal motion during self-motion, assuming the environmental motion remains constant before and during the self-motion (i.e., contextual vision).

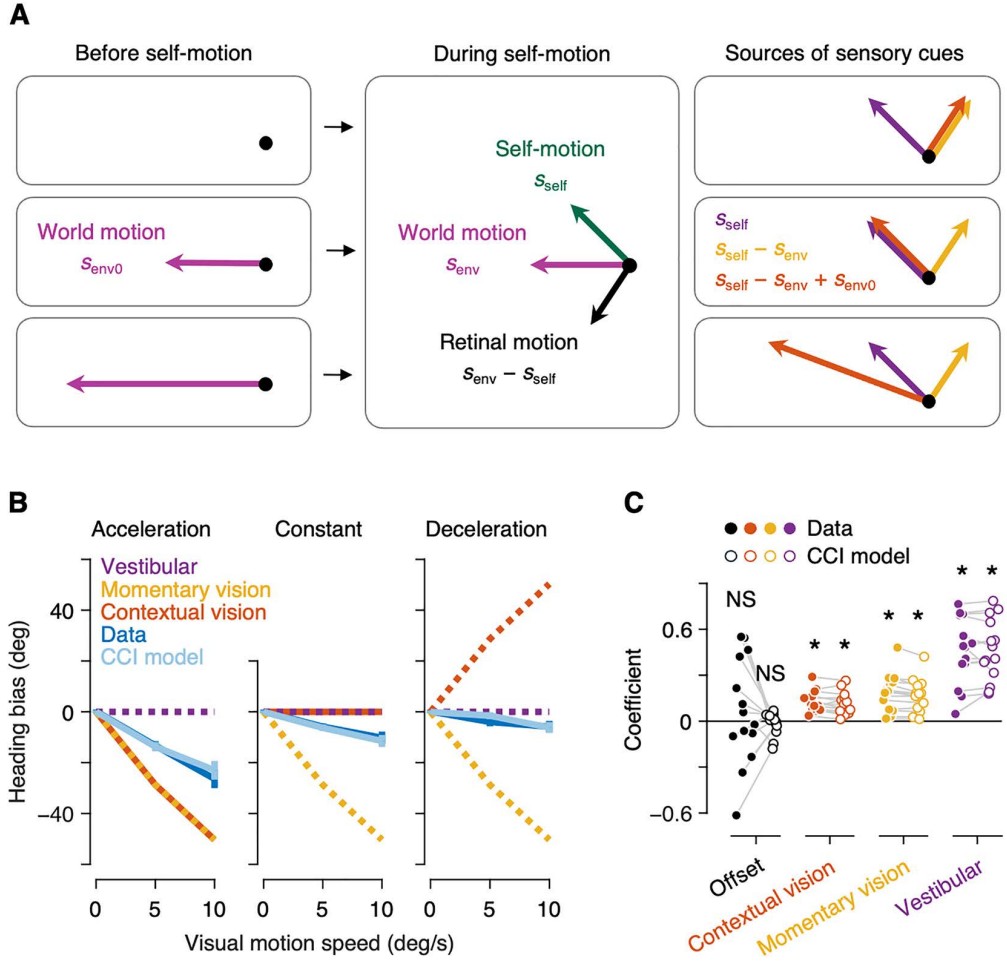

**Fig 3. Heading bias depends on environmental motion presented before self-motion.** (A) Schematic illustration of three sources of sensory cues for Acceleration (top), Constant (middle) and Deceleration (bottom) conditions. (B) Heading bias as a function of visual motion speed across three conditions (i.e., Acceleration, Constant and Deceleration). A negative heading bias indicates that observers' heading estimates are biased in the direction opposite to the visual motion stimuli. Dotted lines represent the predicted heading bias when using vestibular (purple), momentary vision (yellow) and contextual vision (red) exclusively. Solid lines represent human (dark blue) and the Contextual Causal Inference (CCI) model (light blue) observers' heading bias. Error bars represent the standard error of the mean. (C) Linear regression coefficients for human (filled dots) and the CCI model (open dots) observers. *$P < 0.05$, NS $P > 0.05$.

Relying solely on the contextual vision would result in heading estimates centered around the momentary vision plus environmental motion before self-motion, $s_{self} - s_{env} + s_{env0}$. Because the visual motion stimuli presented before self-motion varied across the three visual motion conditions, the heading bias would depend on the visual motion condition (Fig 3B, red). To quantify the influence of different sources of sensory information on observers' heading estimation behavior, we compared the predicted heading bias for each sensory information with the observed heading bias in human data and found that human observers did not rely exclusively on any of the three information. Their heading bias cannot be explained by a combination of any pair of information, either. Instead, it appears that observers utilized a combination of all three sources of information, as evidenced by significant coefficients of linear regression analysis (contextual vision: $t_{13} = 6.920$, $P < 0.001$; momentary vision: $t_{13} = 5.704$, $P < 0.001$; vestibular: $t_{13} = 7.890$, $P < 0.001$; Fig 3C, filled dots).

## Contextual causal inference provides a normative account of the heading estimation behavior

To account for the observed pattern of data, we developed a Contextual Causal Inference (CCI) model that considers two plausible scenarios [24–26]. The first scenario assumes that environmental motion has remained constant before and during the self-motion (i.e., $C = 1$), whereas the second scenario assumes that it has changed (i.e., $C = 2$). To make an optimal estimate, the model observer combines two estimates, each based on a different scenario, weighted by their corresponding posterior probabilities:

$$\hat{s}_{\text{self}} = p\left(C = 1 | x_{\text{vest}}, x_{\text{vis}}, x_{\text{vis0}}\right) \hat{s}_{\text{self},C=1} + p\left(C = 2 | x_{\text{vest}}, x_{\text{vis}}, x_{\text{vis0}}\right) \hat{s}_{\text{self},C=2} \tag{1}$$

where $\hat{s}_{\text{self}}$ is the self-motion estimate, $x_{\text{vest}}$, $x_{\text{vis}}$ and $x_{\text{vis0}}$ are noisy sensory cues the model observer obtained before ($x_{\text{vis0}}$) and during ($x_{\text{vest}}$ and $x_{\text{vis}}$) self-motion, and $\hat{s}_{\text{self},C=1}$ and $\hat{s}_{\text{self},C=2}$ are the self-motion estimate the model observer would make if the environmental motion was perceived constant or different before and during self-motion, respectively. We found that the optimal estimate that minimizes the posterior expected loss in each scenario is given by:

$$\hat{s}_{\text{self},C} = \frac{\frac{x_{\text{vest}}}{\sigma^2_{\text{vest}}} + \frac{\mu_K}{\sigma^2_K} + \frac{0}{\sigma^2_{\text{self}}}}{\frac{1}{\sigma^2_{\text{vest}}} + \frac{1}{\sigma^2_K} + \frac{1}{\sigma^2_{\text{self}}}} \tag{2}$$

where

$$\mu_K = \begin{cases} x_{\text{vis}} - \dfrac{\sigma^2_{\text{env}}}{\sigma^2_{\text{env}} + \sigma^2_{\text{vis0}}} x_{\text{vis0}}, & C = 1 \\ x_{\text{vis}}, & C = 2 \end{cases} \tag{3}$$

$$\sigma^2_K = \begin{cases} \sigma^2_{\text{vis}} + \dfrac{\sigma^2_{\text{env}} \sigma^2_{\text{vis0}}}{\sigma^2_{\text{env}} + \sigma^2_{\text{vis0}}}, & C = 1 \\ \sigma^2_{\text{vis}} + \sigma^2_{\text{env}}, & C = 2 \end{cases} \tag{4}$$

Here, $\sigma^2_{\text{self}}$ and $\sigma^2_{\text{env}}$ are the variance of prior distribution of self-motion, $s_{\text{self}}$, and environmental motion, $s_{\text{env}}$ and $s_{\text{env0}}$, respectively, and $\sigma^2_{\text{vest}}$, $\sigma^2_{\text{vis}}$ and $\sigma^2_{\text{vis0}}$ are the variance of measurement distribution of $x_{\text{vest}}$, $x_{\text{vis}}$ and $x_{\text{vis0}}$, centered around $s_{\text{self}}$, $s_{\text{self}} - s_{\text{env}}$ and $-s_{\text{env0}}$, respectively (see **Methods** for the full description of the generative process and the derivation of the optimal estimates). What stands out from the equations is that both $\hat{s}_{\text{self},C=1}$ and $\hat{s}_{\text{self},C=2}$ are an optimal integration of three sources of information: vestibular, visual and prior information (Eq 2). The crucial distinction between them lies in how visual signals contribute to the inference of self-motion, as characterized by $\mu_K$ and $\sigma^2_K$. Specifically, $\hat{s}_{\text{self},C=1}$ incorporates contextual vision by subtracting the visual signal obtained before self-motion, $x_{\text{vis0}}$, from the one obtained during self-motion, $x_{\text{vis}}$, whereas $\hat{s}_{\text{self},C=2}$ incorporates momentary vision that only reflects the visual signal obtained during self-motion, $x_{\text{vis}}$ (Eq 3). Consequently, the unified estimate (Eq 1) combines contextual vision, momentary vision and vestibular information altogether. Note that we did not predefine these computations. Instead, these computations, including the subtraction, naturally emerged as an optimal solution that minimizes the posterior expected loss (**Methods**).

Nevertheless, the computations performed by the model observer are functionally relevant for each scenario. If environmental motion has remained constant before and during self-motion, it is reasonable for the observer to subtract the posterior estimate of environmental motion, inferred from retinal motion signal collected before self-motion, from the one collected during self-motion and interpret the remaining motion signal as pertaining to self-motion. In contrast, if environmental motion has changed, the observer should disregard retinal motion signal collected before self-motion and instead

rely solely on sensory signals collected during self-motion. Without any evidence indicating otherwise, it is reasonable to assume that the environment is stationary [27–29]. Therefore, the observer assumes that environmental motion during self-motion is close to zero, and that any motion detected on the retina during self-motion is generated entirely by self-motion. This is captured mathematically by momentary vision, with its uncertainty including not only sensory noise but also prior uncertainty about environmental motion.

The fitting results showed that the CCI model provides an excellent fit to the psychophysical data, with average $R^2$ and SEM of $0.714 \pm 0.017$. The model successfully reproduced the characteristic features of human behavior, in that the model observers' heading bias depends not only on visual motion speed but also on visual motion condition (Fig 3B, light blue). We performed the same linear regression analysis as for human data. All three coefficients were significant (contextual vision: $t_{13} = 5.716$, $P < 0.001$; momentary vision: $t_{13} = 5.848$, $P < 0.001$; vestibular: $t_{13} = 8.633$, $P < 0.001$; Fig 3C, open dots) and closely matched those of human observers (Fig 4B). These results indicate that contextual causal inference plays a key role in integrating sensory information over time and has a profound effect on heading perception in a non-stationary environment.

## Alternative models

We compared the performance of the CCI model with that of other alternative models (Fig 4). We first quantitatively compared the model performance using the Akaike information criterion (AIC) to account for differences in model complexity (Fig 4A), with results reported below as the mean AIC difference from the CCI model followed by a bootstrapped 95% confidence interval in brackets. We also compared human and model observers' heading biases (Fig 4C) and linear regression coefficients (Fig 4D) to demonstrate that the CCI model better accounts for the psychophysical data for the majority of the observers.

First, we fit a Momentary Causal Inference (MCI) model, a conventional model of causal inference in multisensory heading perception that does not consider environmental motion before self-motion [16–18]. As expected, unlike human observers, model observers' heading biases did not depend on the visual motion condition. Linear regression coefficients were also drastically different between human and model observers, with model observers' coefficient for contextual vision clustered around zero. Consequently, the MCI model provided a quantitatively worse fit than the CCI model (142.9 [90.6 201.9]).

Next, we considered two special cases of the CCI model. At one extreme, observers may believe that the motion in the environment is constant, even when it is not. Retinal motion during self-motion would then be always subtracted by retinal motion before self-motion to incorrectly infer self-motion from visual signals. Alternatively, observers may believe that the motion in the environment is always independent before and during self-motion. Visual signals collected before self-motion would then be always disregarded. We formulized these strategies in an Integration (Int) model and Segregation (Seg) model, respectively, and found that they cannot reproduce the observed pattern of heading biases. Consequently, both the Int model (256.9 [216.9 299.1]) and the Seg model (112.5 [68.1 162.7]) provided a quantitatively worse fit than the CCI model.

These two alternative models do not consider all available sensory cues; they consider either momentary or contextual vision, along with the vestibular cues. We also considered a Covariance (Cov) model where the observer believes that environmental motion before and during self-motion covary with each other [30–34]. Such temporal correlation leads to a conditional prior that the observer can use to infer the environmental motion during self-motion, after observing the environmental motion before self-motion. Consequently, the model observer performs a linear integration of all available sensory signals, with the temporal correlation of the environmental motion determining the contribution of visual signal collected before self-motion. While the Cov model exhibited a qualitatively more similar pattern to the data than the above alternative models, it provided a significantly worse fit than the CCI model (28.3 [5.1 54.1]), with 10 out of 14 observers' data favoring the CCI model. We similarly considered a Fixed Weight (Fix) model, a rather descriptive model that

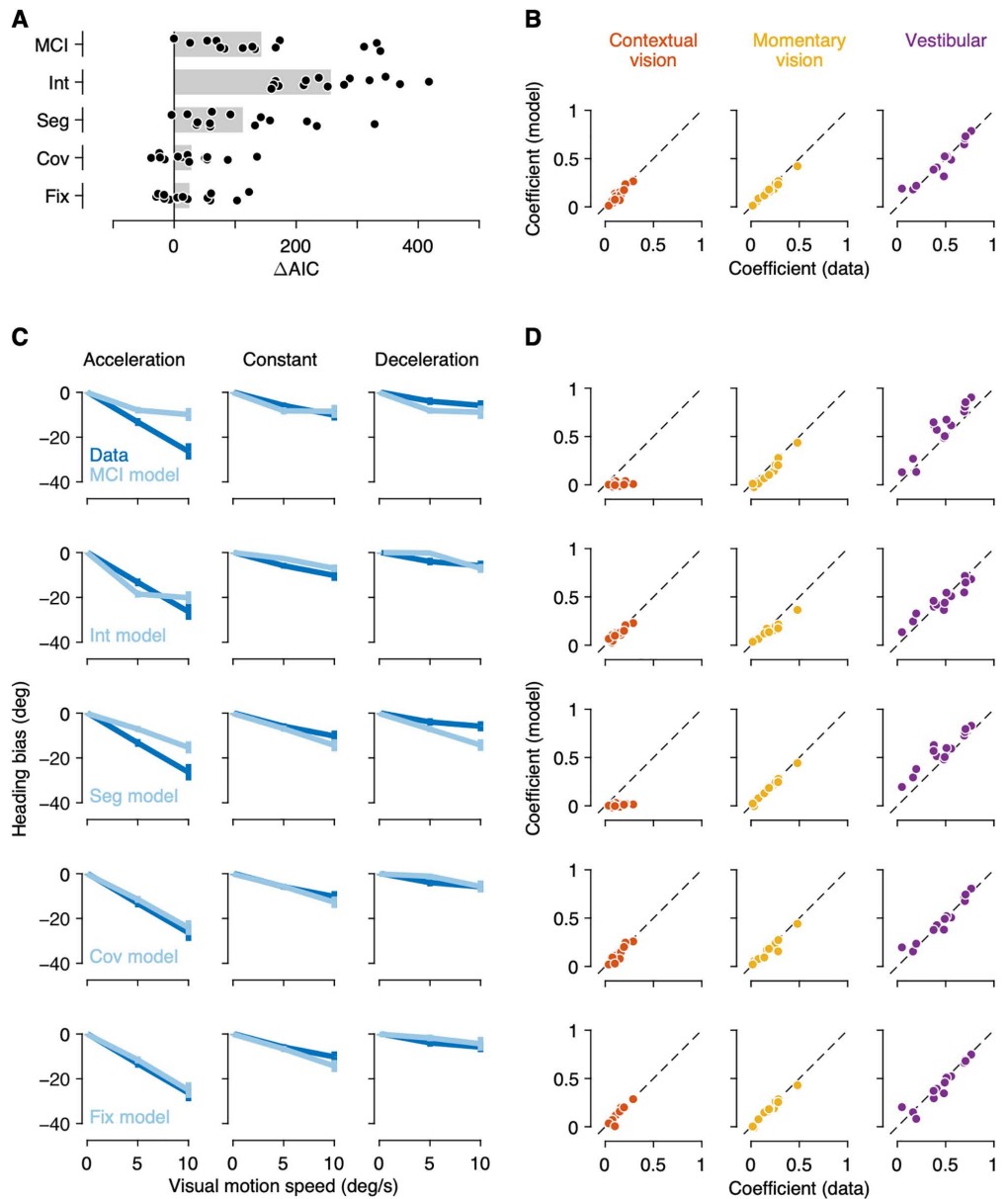

**Fig 4. Alternative models.** (A) Comparison of goodness-of-fit among observer models. Small dots represent individual observers' difference in Akaike information criterion (AIC) values between each model and the CCI model, and gray bars represent the average. Positive values indicate a worse fit than the CCI model. (B) Comparison of linear regression coefficients between human and the CCI model observers. (C) Same as in Fig 3A but for alternative models. (D) Same as in B but for alternative models.

computes a weighted sum of all available sensory signals with fixed weights, and obtained similar results (25.0 [2.2 49.9]), with 9 observers' data favoring the CCI model.

A key prediction of the CCI model is that the heading estimate is a nonlinear integration of sensory signals, because the weight is determined by the observer's inference about whether environmental motion has remained constant. That is, the observer would adaptively adjust the weight depending on the available sensory signals. To test this prediction, we analyzed human observers' weight on the heading estimate assuming constant motion in the environment, conditioned upon

whether it was actually constant. If observers performed a linear integration of sensory signals, the weight would be the same across all trials (Fig 5A, left). However, if observers performed causal inference, the weight on the heading estimate assuming constant motion in the environment would be larger when it was actually constant (Fig 5A, right). We found that human observers indeed employed adaptive weights to integrate sensory signals, consistent with the causal inference prediction ($t_{13} = 3.754$, $P = 0.002$; Fig 5B, left). Applying the same analysis on the model observers' behavior, we confirmed that the CCI model also used adaptive weights ($t_{13} = 4.615$, $P < 0.001$; Fig 5B, right).

The results that the observers performed a contextual causal inference does not necessarily mean they did it optimally. To test this, we fit two variants of causal inference model: a Heuristic model that performs the causal inference without taking into account sensory uncertainty, and a Winner-Take-All model that commits entirely to the more probable scenario without taking into account the less probable one. Both the Heuristic model (64.7 [39.5 93.1]) and the Winner-Take-All model (39.7 [20.3 60.5]) showed a significantly worse fit than the CCI model, suggesting that the observers performed the contextual causal inference optimally by taking into account sensory uncertainty [35].

## Discussion

A moving observer faces an interpretational challenge if the surrounding environment is also in motion, because motion on the retina could be attributed to movement in the environment, to movement of the observer or to some combination of the two. In this study, we asked whether observers rely on temporal context in heading perception. Specifically, we reasoned that observers would consider what they observed before they begin to move when estimating the current heading direction. By manipulating visual motion stimuli before self-motion while controlling for the visual motion stimuli during self-motion, we demonstrated that environmental motion that occurred ahead of self-motion indeed systematically influences heading direction estimations. We also tested whether a causal inference scheme can account for the observed pattern of behavior and provided a normative explanation about whether and how observers integrate sensory information about self-motion obtained across time.

Most previous work has studied multisensory heading perception in a temporally isolated context [8–23]. They focused on the integration of visual and vestibular signals acquired during self-motion and found that observers use the vestibular signal to parse out motion in the environment from motion on the retina. While these and related studies provide valuable insights into multisensory integration, causal inference and neural correlates of heading perception, they could not, by design, examine whether heading perception depends on the temporal context. Going beyond the conventional approach,

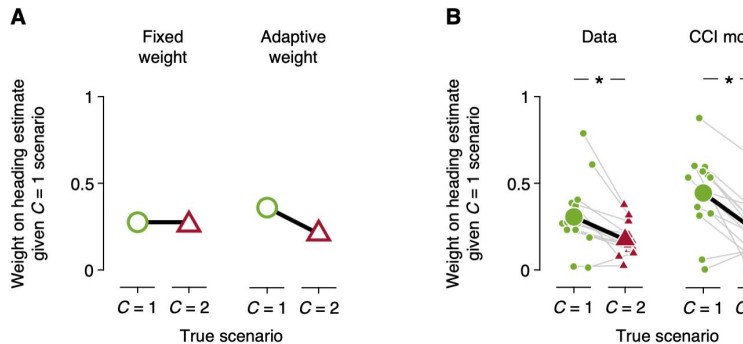

**Fig 5. Adaptive weight as a signature of causal inference.** (A) Prediction about weights on the heading estimate associated with constant motion in the environment. For the linear sensory integration with fixed weights (left), the weights would be the same. For the nonlinear integration with adaptive weights (right), the weights would be larger when the environmental motion was constant (i.e., $C = 1$). (B) Weights on the heading estimate associated with constant motion in the environment for human (left) and the CCI model (right) observers. Small markers connected by gray lines represent weights for individual observers, and big markers connected by a black line represent the corresponding averages. *$P < 0.05$.

we introduced a temporal component in the experiment and showed that observers integrate visual signals acquired before self-motion with visual and vestibular signals acquired during self-motion.

We have shown that the perception of heading direction in a non-stationary environment can be well understood under a causal inference framework. The causal inference framework [36–38] has been widely employed to study the computational mechanism underlying visual motion perception [33,39,40] and multisensory heading perception [16–18]. Its neural correlates have been also actively investigated [23,41–49]. Inspired by a previous work that used the causal inference framework to understand heading perception in the presence of object motion [9], we extended the causal inference framework to temporal domain. We showed whether and how an ideal observer should integrate information about self-motion obtained across time, if they believe that environmental motion has remained constant or changed before and during self-motion. Our findings suggest that causal inference operates across temporal domain and that the interactions between environmental motion before and during self-motion are governed by normative principles.

Our causal inference model indicates that observers use their prior belief about environmental motion to estimate self-motion. Although its specific functional form differs across studies [27,39,40], it is generally believed that human observers use a speed prior centered on zero [28]. In our model, when environmental motion has remained constant before and during self-motion, observers use the retinal motion signal collected before self-motion to parse out the environmental motion from the retinal motion signal collected during self-motion. On the other hand, when environmental motion has changed, the observers are left with no other information about environmental motion except for their prior belief. Therefore, observers assume, with much uncertainty, that the environment is stationary, thereby attributing retinal motion entirely to self-motion. Our model is consistent with previous work that showed how the slow-speed prior influences heading perception in the presence of object motion [9]. Similar to our model, their model relies on the prior belief that objects in the environment tend to be stationary, attributing retinal motion of the object entirely to self-motion, but with increased uncertainty.

It is well established that sensory signals should only be integrated when they are close in time, space, and content [50–64], which has become the crux of causal inference in perception [24–26]. In this study, we showed that heading estimation is governed by the same principle. For example, when contextual vision assuming constant motion in the environment specifies a heading direction very different from the heading direction suggested by vestibular signal, an ideal observer determines that the most likely interpretation is that environmental motion has not remained constant. As a result, the observer adaptively weighs more on the interpretation assuming the constant motion in the environment when it was actually constant (Fig 5). This is in stark contrast with the prediction of any model that linearly integrates sensory signals, for example, based on a two-dimensional prior with a non-zero correlation [65]. One may extend such a model to have heavy-tailed conditional priors [60]. Indeed, recent studies have explained a nonlinear sensory integration across time by modeling a heavy-tailed conditional prior [66,67], consistent with the statistics of the natural environment [67–71]. While an earlier study has argued that a causal inference model that explicitly considers two causal structures can better explain human behavior than the model with heavy-tailed conditional priors [25], it has been later shown that those two models are mathematically compatible and produce similar nonlinear patterns [24,37]. One advantage of modeling the causal structure as a random variable is that it becomes straightforward to explain how observers make a causality judgment [25]. While we did not collect explicit causality judgments in this study, such data may help constrain the model fit.

While our study involves passive movements of the observer, it would be interesting to combine our paradigm with volitional movements to see how motor information about heading direction is integrated as well [5]. When an observer makes a volitional movement, the motor system generates an internal copy of the motor signal. This efference copy can be collated with the reafferent sensory signal that results from the observer's movement, enabling a comparison of actual movement with desired movement [72]. That is, observers making a volitional movement are able to predict what they should see if they moved as intended. Note that even with this additional source of information, observers still need to consider environmental motion presented ahead of self-motion, since it provides the context with which observers make

predictions about what they would see. A possible prediction is that observers could better disambiguate motion on the retina during active rather than passive self-motion [22], leading to more accurate heading perception, but the specific mechanism by which observers integrate uncertain information is still an open question.

Considering the rich history of computational studies with clinical implications [73–76], our findings may have strong clinical implications to balance disorders and falls. Falls are the leading cause of accidental injury and death, especially among older adults [77]. As balance control is achieved based on self-motion detected by visual, vestibular and proprioceptive sensory systems [78], understanding the computational mechanism underlying the multisensory integration of information about self-motion is crucial to clinical research on balance disorders. Furthermore, as demonstrated in the classical study where young stationary observers fell in response to the visual motion of a whole scene [79], balance control is strongly influenced by visual motion signals [77], also a key component addressed in this study. Therefore, we believe that our findings can be used to further characterize behavior of special populations such as older adults and patients with balance disorders and may become a steppingstone in understanding falls and in the development of prevention strategies.

The stimulus used in our study—in which the entire visible environment moves uniformly—is uncommon in everyday life. Aside from the example introduced in the Introduction, where a large bus fills most of the visual field, rare instances include situations where observers inside a train view the outside world through an aperture entirely filled by the motion of another train. In typical daily visual experiences, the visual field consists mostly of stationary objects and backgrounds, with only occasional moving elements. This simplification of the experimental setup is a limitation of the current study; however, it also enables a clearer identification of the targeted mechanism: the effects of synchrony between visual and vestibular signals.

Another related factor not directly addressed in our study is the visual–vestibular temporal binding window—the time interval within which visual and vestibular cues are perceptually integrated as originating from the same event. The temporal binding window varies depending on modality, stimulus properties and individual sensitivity, ranging from as narrow as 15 ms to as broad as 400 ms [80]. More recent studies showed that visual motion can bias inertial heading judgments when the onset difference is 250 ms or less, but this influence diminishes beyond 500 ms and disappears entirely at a 1,000 ms delay [81]. In our design, the visual motion preceded self-motion by 2 seconds, placing the two cues well outside the typical temporal binding window. Thus, multisensory integration via temporal binding was unlikely to contribute to the observed effects. Instead, it is more plausible that the earlier environmental motion was subtracted from the overall retinal motion to estimate the self-motion. A subtraction of motion vectors is also performed in optic flow parsing but in the opposite way—the brain estimates environmental motion by subtracting self-motion from the overall retinal motion pattern [10,11,13,14,82–85]. While the concept of the temporal binding window remains central to multisensory integration, it was not a contributing factor in the current design and instead represents an important direction for future investigation. Accordingly, future research could broaden the study's scope by examining how synchrony interacts with optic flow parsing when only parts of the visual environment are in motion.

In summary, we demonstrated that human observers moving in a non-stationary environment use visual motion signal collected before self-motion to estimate their current heading direction. Our results suggest that the brain interprets ambiguous retinal motion signal using a normative causal inference framework, opening a new avenue for future work regarding neural correlates of heading perception and multisensory integration across time. Our findings could be informative for future research on behavior and pathological conditions affecting balance control.

## Methods

### Ethics statement

We obtained written informed consent prior to the experiment. All procedures were approved by Ulsan National Institute of Science and Technology Institutional Review Board (UNISTIRB-23–004-A).

## Observers

Fifteen young adults (6 female, age: 19 – 27 years) participated in this study. We excluded one observer's data from analysis, as this observer always reported ±30° throughout the experiment, presumably misunderstanding the task instruction. All observers were naïve as to the purpose of the experiment and reported normal or corrected-to-normal vision.

## Apparatus

Experimental setup is shown in Fig 1B. Observers sat comfortably on a car seat mounted on a six-degree-of-freedom motion platform (CKAS 6DOF Motion Systems, CKAS Mechatronics Pty Ltd) and leaned their forehead and chin on a cushioned chinrest also mounted on the motion platform, thereby immobilizing their head relative to the platform. The trajectory of the motion platform was controlled in real time at 60 Hz. A DLP projector (DepthQ WXGA 360, LightSpeed Design Inc) with a pixel resolution of 1280 × 720 back-projected visual images onto a large screen (211 cm × 119 cm) at 60 Hz. Both the projector and the screen were placed outside the motion platform. Observers viewed images on the screen through a custom-built 55-cm-diameter circular aperture that was rigidly mounted on the motion platform. The distance between the screen and the aperture was 43 cm, and the distance between the aperture and the observer was 60 cm, resulting in a visual field with a radius of 25°. To prevent access to extraneous visual cues outside the intended visual field, opaque side panels were integrated around the headrest. These panels acted as peripheral occluders, ensuring that participants could only see the visual stimuli presented through the aperture and were shielded from any surrounding environmental cues. To ensure precise temporal alignment between the visual and vestibular stimuli, we implemented a calibration procedure based on empirically measured motion platform delay, obtained using the built-in encoder system of the platform (S1 Supporting information).

## Stimuli

Inertial (vestibular) motion stimuli were delivered via the motion platform that transported observers for 2 s in one of ten directions in the frontal plane: ±5°, ±15°, ±25°, ±35° or ±45° relative to vertically upward. The motion followed a modified raised cosine velocity profile, with a peak velocity of 8.5°/s and peak acceleration and deceleration of ±0.167°/s². Unlike a conventional raised cosine velocity profile, our profile held the peak velocity of 8.5°/s for 0.4 s before decelerating back to rest at 0°/s (Fig 1C, green).

Visual stimuli were generated using the Psychophysics Toolbox [86] in MATLAB and projected onto the rear-projection screen via the projector positioned behind the screen. To ensure that only the intended visual stimuli were presented without any additional visual signals, the size of the visual stimuli was precisely adjusted to cover the entire area of the physical aperture through which the visual stimuli were presented. The visual stimuli were non-rigid texture motion generated by bandpass filtering white noise with a Gaussian envelope in the coordinates of speed, frequency and orientation [87–91]. The Gaussian envelope was fully characterized by its mean and bandwidth (i.e., the standard deviation). Specifically, a given image $I$ is defined by:

$$I = F^{-1}\left(\exp\left(-\frac{1}{2}\left(\frac{f_x V_x + f_y V_y + f_t}{B_v f_r}\right)^2\right) \cdot \frac{1}{f_r}\exp\left(-\frac{1}{2}\left(\frac{\ln\left(\frac{f_r}{sf_0}\right)}{\ln\left(\frac{sf_0 + B_{sf}}{sf_0}\right)}\right)^2\right) \cdot \exp(i\Phi)\right)$$

(5)

where $F^{-1}(\cdot)$ denotes an inverse Fourier transform, $V = (V_x, V_y)$ the central speed, $f_r = \sqrt{f_x^2 + f_y^2}$ the radial frequency and $\Phi$ a uniformly distributed phase spectrum in $[0, 2\pi)$. The visual stimuli only moved either leftward or rightward, with $V_y = 0$, and $V_x$ following a modified raised cosine velocity profile (see below). The speed bandwidth $B_v$ was set to 2.1°/s, resulting in non-rigid motion that constantly changes its form over time. We defined the Gaussian envelope on a logarithmic frequency scale [92,93], setting the central spatial frequency $sf_0$ and the spatial frequency bandwidth $B_{sf}$ to 0.5 cpd. All

orientations were equally selected, yielding a toroidal envelope. Finally, the envelope was used to linearly filter a white-noise stimulus drawn from a uniform distribution. Three example movies are shown (S1–S3 Movies) for the visual speed of 0°/s, 5°/s and 10°/s, respectively.

Observers experienced three visual motion conditions during the experiment (Fig 1C, magenta). In all conditions, visual motion had five desired velocities during inertial motion: 0°/s, ±5°/s and ±10°/s. For each desired velocity, visual motion presented before inertial motion varied systematically by condition. (1) In Constant condition, the visual motion velocity remained constant at the desired velocity before and during inertial motion. (2) In Acceleration condition, the visual motion velocity started at 0°/s and then accelerated to the desired velocity with the onset of inertial motion. (3) In Deceleration condition, the visual motion velocity was initially set at twice the desired velocity and then decelerated to the desired velocity with the onset of inertial motion. In both Acceleration and Deceleration conditions, after the velocity reached its desired velocity, it was maintained for 0.4 s and then gradually returned to the initial velocity over 0.8 s, following the modified raised cosine velocity profile.

### Task and procedure

The sequence of events on a trial is shown in Fig 1D. Observers sat on the motion platform and binocularly viewed a visual motion stimulus through the aperture for 2 s. They were then passively moved upward for 2 s in one of ten directions, while continuing to view the visual motion stimulus. Following the synchronized offset of visual and inertial motion stimuli, an adjustable probe appeared on the screen, and observers reported the perceived direction of self-motion by adjusting the probe using a computer mouse. Trials were separated by a 2.85-s inter-trial interval during which the screen was blank and the motion platform moved back to its initial location. There was a total of 150 distinct stimulus conditions (5 visual motion velocities × 10 inertial motion directions × 3 visual motion conditions), and each condition was repeated five times, resulting in a total of 750 trials. The three visual motion conditions, each defined by different prior temporal dynamics, were randomly interleaved within each session to prevent predictability and ensure proper counterbalancing. The experiment was conducted over two sessions, with each session consisting of 15 blocks of 25 trials each.

### Data analysis

To assess the observers' perceptual performance, we calculated their heading bias as the average angular difference between the perceived heading direction reported by the observers and the true heading direction, with errors realigned such that positive deviations are in the direction of the visual motion stimuli. Thus, a negative heading bias indicates that the heading estimates are biased in the direction opposite to the visual motion stimuli in the frontal plane. Our analysis leveraged the symmetry of the heading bias by collapsing the bias onto one side of the graph. This was achieved by mirroring each observer's heading bias in the origin, expressing it as a function of the speed, not velocity, of visual motion stimuli. That is, we flipped the sign of the heading bias on the trials with the leftward (i.e., negative) visual motion, and plotted the heading bias on all trials as a function of unsigned visual motion velocity (i.e., visual motion speed). A repeated-measures ANOVA with two factors (visual motion speed: 0°/s, 5°/s and 10°/s; visual motion condition: Acceleration, Constant and Deceleration) was performed on the observers' heading bias.

We considered three sources of sensory information about the heading direction (Fig 3A). First, observers only using the vestibular information would veridically report the direction of inertial motion. Second, observers only using momentary vision would report the direction of inertial motion subtracted by visual motion during inertial motion. Lastly, observers using the contextual vision would report the direction of inertial motion subtracted by visual motion during inertial motion and added by visual motion before inertial motion. To quantitatively assess contributions of each source of information, we fit a linear regression model with the observers' heading estimate as a dependent variable and the predicted heading bias for each source of information as independent variables:

$$\hat{s}_{\text{self}} = \beta_0 + \beta_1 s_{\text{self}} + \beta_2 \left( s_{\text{self}} - s_{\text{env}} \right) + \beta_3 \left( s_{\text{self}} - s_{\text{env}} + s_{\text{env0}} \right) \tag{6}$$

where $s_{\text{self}}$ represents self-motion and $s_{\text{env0}}$ and $s_{\text{env}}$ represent environmental motion before and during self-motion, respectively (see also below).

### Contextual causal inference (CCI) model

Inspired by a previous work that extended a causal inference framework to heading perception in the presence of object motion [9], we use the causal inference framework to model how an ideal observer infers the heading direction from noisy and ambiguous sensory input when the environment is also in motion. Taking a Bayesian approach, we begin by specifying how task-relevant variables are interrelated statistically. We first define $C \in \{1, 2\}$ as the causal structure of the motion in the environment, indicating whether the environmental motion before and during self-motion is constant (i.e., $C = 1$) or independent (i.e., $C = 2$). We assume that $C$ follows a binomial distribution with $p(C = 1) = p_{\text{constant}}$, which represents the prior probability that the environmental motion remains constant before and during self-motion. We treat $p_{\text{constant}}$ as a free parameter, as it has been shown to be stable within individual observers but not necessarily tied to the statistics of the task [94], at least without a prolonged exposure [95]. We define $s_{\text{self}}$ as the lateral component of the observers' heading in the frontal plane, and $s_{\text{env0}}$ and $s_{\text{env}}$ as the environmental motion in a fronto-parallel plane before and during self-motion, respectively. Motivated by the rich literature on the slow-speed prior [27–29,96], we assume that $s_{\text{self}}$ follows a zero-mean Gaussian prior distribution $\mathcal{N}\left(0, \sigma_{\text{self}}^2\right)$, and $s_{\text{env0}}$ and $s_{\text{env}}$ follow a zero-mean Gaussian prior distribution $\mathcal{N}\left(0, \sigma_{\text{env}}^2\right)$. If the environmental motion before and during self-motion is constant (i.e., $C = 1$), $s_{\text{env0}}$ is drawn from the prior, and $s_{\text{env}}$ takes the same value. If the motion is independent (i.e., $C = 2$), both $s_{\text{env0}}$ and $s_{\text{env}}$ are independently drawn from the same prior. Note that while many previous models have assumed Gaussian priors [9,28,97–99] or Gaussian process priors such as Ornstein-Uhlenbeck process [30–34] for analytical tractability, the specific functional form of the slow-speed prior has been debated, with other models assuming a power-law function [27,29,96,100,101] or mixture priors with a point-mass at zero [39,40]. The structure of our model is compatible with such alternatives, though extensions may require additional free parameters and/or numerical rather than analytic solutions even in early stages of the derivation [16].

The observer does not have access to the true direction of self-motion. Instead, the observer only has access to visual and vestibular cues to self-motion. The motion of the environment before self-motion provides a noisy visual signal, $x_{\text{vis0}}$, which follows a Gaussian measurement distribution $\mathcal{N}\left(-s_{\text{env0}}, \sigma_{\text{vis0}}^2\right)$. The motion of the environment during self-motion as viewed by the observer's eye is a combination of the environmental motion, $s_{\text{env}}$, and the self-motion, $s_{\text{self}}$. Thus, $x_{\text{vis}}$ represents a noisy visual signal collected during self-motion and follows a Gaussian measurement distribution $\mathcal{N}\left(s_{\text{self}} - s_{\text{env}}, \sigma_{\text{vis}}^2\right)$. If the environmental motion before and during self-motion is constant, the combination of $x_{\text{vis}}$ and $x_{\text{vis0}}$ can be viewed as a compound cue about self-motion, $s_{\text{self}}$. On the other hand, a vestibular signal, $x_{\text{vest}}$, is generated solely from self-motion. Specifically, $x_{\text{vest}}$ represents a noisy vestibular signal about self-motion and follows a Gaussian measurement distribution $\mathcal{N}\left(s_{\text{self}}, \sigma_{\text{vest}}^2\right)$. For $x_{\text{vis0}}$ and $x_{\text{vis}}$, we assume that the sensory noise follows Weber's law [102–105], i.e., $\sigma_{\text{vis0}}^2 = w_{\text{vis}}^2 \left| -s_{\text{env0}} \right|^2$ and $\sigma_{\text{vis}}^2 = w_{\text{vis}}^2 \left| s_{\text{self}} - s_{\text{env}} \right|^2$, where $w_{\text{vis}}$ is a constant Weber fraction.

With this generative model in mind, the observer infers the direction of self-motion from noisy and ambiguous sensory signals. To do so, the observer first constructs the posterior probability distribution over self-motion, $p\left(s_{\text{self}} | x_{\text{vest}}, x_{\text{vis}}, x_{\text{vis0}}\right)$, which represents the observer's belief about self-motion, $s_{\text{self}}$, after receiving the sensory signals, $x_{\text{vest}}$, $x_{\text{vis}}$ and $x_{\text{vis0}}$. From this posterior, the observer computes a single estimate, $\hat{s}_{\text{self}}$, by considering a loss function, $\ell\left(\hat{s}_{\text{self}}, s_{\text{self}}\right)$, which quantifies the cost of erroneously estimating $s_{\text{self}}$ as $\hat{s}_{\text{self}}$. The Bayesian estimate is the one that minimizes the posterior expected loss, which is the expected value of the loss function under the posterior distribution over $s_{\text{self}}$.

$$\hat{s}_{\text{self}} = \underset{\hat{s}_{\text{self}}}{\arg\min} \int \ell\left(\hat{s}_{\text{self}}, s_{\text{self}}\right) p\left(s_{\text{self}} | x_{\text{vest}}, x_{\text{vis}}, x_{\text{vis0}}\right) ds_{\text{self}} \tag{7}$$

Since the generative model depends on the latent causal structure, $C$, of the motion in the environment, inference about self-motion must also depend on $C$. However, the observer does not have access to the true causal structure. We therefore rewrite the posterior of self-motion as a marginalization over all possible causal structures:

$$p\left(s_{\text{self}}|x_{\text{vest}}, x_{\text{vis}}, x_{\text{vis0}}\right) = \sum_C p\left(s_{\text{self}}|x_{\text{vest}}, x_{\text{vis}}, x_{\text{vis0}}, C\right) p\left(C|x_{\text{vest}}, x_{\text{vis}}, x_{\text{vis0}}\right) \tag{8}$$

Assuming a squared error loss, $\updownarrow(\hat{s}_{\text{self}}, s_{\text{self}}) = (\hat{s}_{\text{self}} - s_{\text{self}})^2$ [106,107], the optimal estimate becomes the mean of the posterior. Substituting the marginalized posterior into the estimator, we obtain:

$$\hat{s}_{\text{self}} = \int s_{\text{self}}\, p\left(s_{\text{self}}|x_{\text{vest}}, x_{\text{vis}}, x_{\text{vis0}}\right) ds_{\text{self}} = \sum_C p\left(C|x_{\text{vest}}, x_{\text{vis}}, x_{\text{vis0}}\right) \int s_{\text{self}}\, p\left(s_{\text{self}}|x_{\text{vest}}, x_{\text{vis}}, x_{\text{vis0}}, C\right) ds_{\text{self}} \tag{9}$$

Thus, the optimal heading estimate becomes a combination of two optimal heading estimates assuming each causal structure $C$, weighted by its posterior probability (i.e., model averaging [108]). To compute the optimal heading estimate, the observer has to infer whether environmental motion has remained constant, and simultaneously compute the optimal heading estimates assuming that environmental motion has remained constant or has changed.

The inference of whether environmental motion has remained constant before and during self-motion is performed optimally using Bayes' rule:

$$p\left(C|x_{\text{vest}}, x_{\text{vis}}, x_{\text{vis0}}\right) = \frac{p(x_{\text{vest}}, x_{\text{vis}}, x_{\text{vis0}}|C)p(C)}{p(x_{\text{vest}}, x_{\text{vis}}, x_{\text{vis0}})} \tag{10}$$

The posterior probability that the environmental motion has remained constant is given by:

$$p\left(C=1|x_{\text{vest}}, x_{\text{vis}}, x_{\text{vis0}}\right) = \frac{p(x_{\text{vest}}, x_{\text{vis}}, x_{\text{vis0}}|C=1)p_{\text{constant}}}{p(x_{\text{vest}}, x_{\text{vis}}, x_{\text{vis0}}|C=1)p_{\text{constant}} + p(x_{\text{vest}}, x_{\text{vis}}, x_{\text{vis0}}|C=2)(1-p_{\text{constant}})} \tag{11}$$

Analogously, the posterior probability that the environmental motion has changed is given by:

$$p\left(C=2|x_{\text{vest}}, x_{\text{vis}}, x_{\text{vis0}}\right) = \frac{p(x_{\text{vest}}, x_{\text{vis}}, x_{\text{vis0}}|C=2)(1-p_{\text{constant}})}{p(x_{\text{vest}}, x_{\text{vis}}, x_{\text{vis0}}|C=1)p_{\text{constant}} + p(x_{\text{vest}}, x_{\text{vis}}, x_{\text{vis0}}|C=2)(1-p_{\text{constant}})} \tag{12}$$

We rewrite the likelihood by using dependencies of the sensory measurements on the true states. When $C=1$, we can substitute $s_{\text{env0}}$ with $s_{\text{env}}$. In addition, $s_{\text{env}}$ has no bearing on $x_{\text{vest}}$, so we can safely take the conditional probability distribution $p\left(x_{\text{vest}}|s_{\text{self}}\right)$, as well as the prior $p\left(s_{\text{self}}\right)$, out of the integral with respect to $s_{\text{env}}$. As a result, the likelihood can be rewritten as:

$$\begin{aligned} p\left(x_{\text{vest}}, x_{\text{vis}}, x_{\text{vis0}}|C=1\right) &= \iint p\left(x_{\text{vest}}, x_{\text{vis}}, x_{\text{vis0}}|s_{\text{self}}, s_{\text{env}}\right) p\left(s_{\text{self}}, s_{\text{env}}\right) ds_{\text{self}} ds_{\text{env}} \\ &= \iint p\left(x_{\text{vest}}|s_{\text{self}}\right) p\left(x_{\text{vis}}|s_{\text{self}}, s_{\text{env}}\right) p\left(x_{\text{vis0}}|s_{\text{env}}\right) p\left(s_{\text{self}}\right) p\left(s_{\text{env}}\right) ds_{\text{self}} ds_{\text{env}} \\ &= \int p\left(x_{\text{vest}}|s_{\text{self}}\right) \left(\int p\left(x_{\text{vis}}|s_{\text{self}}, s_{\text{env}}\right) p\left(x_{\text{vis0}}|s_{\text{env}}\right) p\left(s_{\text{env}}\right) ds_{\text{env}}\right) p\left(s_{\text{self}}\right) ds_{\text{self}} \end{aligned} \tag{13}$$

When $C=2$, $s_{\text{env0}}$ and $s_{\text{env}}$ are two distinct entities, so we need to deal with a triple integral. Both $x_{\text{vest}}$ and $x_{\text{vis}}$ do not depend on $s_{\text{env0}}$, and $x_{\text{vis0}}$ depends on $s_{\text{env0}}$ alone. Therefore, we can separate out the triple integral into a product of a double integral and a single integral. As a result, the likelihood can be rewritten as:

$$\begin{aligned} p\left(x_{\text{vest}}, x_{\text{vis}}, x_{\text{vis0}}|C=2\right) &= \iiint p\left(x_{\text{vest}}, x_{\text{vis}}, x_{\text{vis0}}|s_{\text{self}}, s_{\text{env}}, s_{\text{env0}}\right) p\left(s_{\text{self}}, s_{\text{env}}, s_{\text{env0}}\right) ds_{\text{self}} ds_{\text{env}} ds_{\text{env0}} \\ &= \iiint p\left(x_{\text{vest}}|s_{\text{self}}\right) p\left(x_{\text{vis}}|s_{\text{self}}, s_{\text{env}}\right) p\left(x_{\text{vis0}}|s_{\text{env0}}\right) p\left(s_{\text{self}}\right) p\left(s_{\text{env}}\right) p\left(s_{\text{env0}}\right) ds_{\text{self}} ds_{\text{env}} ds_{\text{env0}} \\ &= \left(\int p\left(x_{\text{vest}}|s_{\text{self}}\right) \left(\int p\left(x_{\text{vis}}|s_{\text{self}}, s_{\text{env}}\right) p\left(s_{\text{env}}\right) ds_{\text{env}}\right) p\left(s_{\text{self}}\right) ds_{\text{self}}\right) \left(\int p\left(x_{\text{vis0}}|s_{\text{env0}}\right) p\left(s_{\text{env0}}\right) ds_{\text{env0}}\right) \end{aligned} \tag{14}$$

Since all the probability distributions in the integrals in Eqs 13 and 14 are Gaussians, we can analytically solve:

$$p\left(x_{\text{vest}}, x_{\text{vis}}, x_{\text{vis0}} \mid C\right) = \frac{K_0}{2\pi\sqrt{\sigma_{\text{vest}}^2\sigma_{\text{K}}^2 + \sigma_{\text{vest}}^2\sigma_{\text{self}}^2 + \sigma_{\text{K}}^2\sigma_{\text{self}}^2}} \exp\left(-\frac{1}{2}\frac{(x_{\text{vest}}-\mu_{\text{K}})^2\sigma_{\text{self}}^2 + x_{\text{vest}}^2\sigma_{\text{K}}^2 + \mu_{\text{K}}^2\sigma_{\text{vest}}^2}{\sigma_{\text{vest}}^2\sigma_{\text{K}}^2 + \sigma_{\text{vest}}^2\sigma_{\text{self}}^2 + \sigma_{\text{K}}^2\sigma_{\text{self}}^2}\right) \tag{15}$$

where $\mu_{\text{K}}$ and $\sigma_{\text{K}}^2$ are given in Eqs 3 and 4, respectively, and

$$K_0 = \frac{1}{\sqrt{2\pi\left(\sigma_{\text{vis0}}^2 + \sigma_{\text{env}}^2\right)}} \exp\left(-\frac{1}{2}\frac{x_{\text{vis0}}^2}{\sigma_{\text{vis0}}^2 + \sigma_{\text{env}}^2}\right) \tag{16}$$

is cancelled out when calculating Eqs 11 and 12, as it does not depend on $C$.

Having computed the posterior probability of each causal structure (Eq 10), we proceed to compute the heading estimate associated with each causal structure:

$$\hat{s}_{\text{self},C} = \int s_{\text{self}}\, p\left(s_{\text{self}} \mid x_{\text{vest}}, x_{\text{vis}}, x_{\text{vis0}}, C\right) ds_{\text{self}} \tag{17}$$

which, combined with Eq 9, yields Eq 1. The posterior probability of self-motion is proportional to the product of the likelihood multiplied by the prior:

$$p\left(s_{\text{self}} \mid x_{\text{vest}}, x_{\text{vis}}, x_{\text{vis0}}, C\right) \propto p\left(x_{\text{vest}}, x_{\text{vis}}, x_{\text{vis0}} \mid s_{\text{self}}, C\right) p\left(s_{\text{self}}\right) \tag{18}$$

When the environmental motion has remained constant before and during self-motion, $x_{\text{vest}}$, $x_{\text{vis}}$ and $x_{\text{vis0}}$ all provide relevant information about the observer's heading. As in Eq 13, we substitute $s_{\text{env0}}$ with $s_{\text{env}}$. Thus, the likelihood can be rewritten as:

$$\begin{aligned} p\left(x_{\text{vest}}, x_{\text{vis}}, x_{\text{vis0}} \mid s_{\text{self}}, C=1\right) &= \int p\left(x_{\text{vest}}, x_{\text{vis}}, x_{\text{vis0}} \mid s_{\text{self}}, s_{\text{env}}\right) p\left(s_{\text{env}}\right) ds_{\text{env}} \\ &= \int p\left(x_{\text{vest}} \mid s_{\text{self}}\right) p\left(x_{\text{vis}} \mid s_{\text{self}}, s_{\text{env}}\right) p\left(x_{\text{vis0}} \mid s_{\text{env}}\right) p\left(s_{\text{env}}\right) ds_{\text{env}} \\ &= p\left(x_{\text{vest}} \mid s_{\text{self}}\right)\left(\int p\left(x_{\text{vis}} \mid s_{\text{self}}, s_{\text{env}}\right) p\left(x_{\text{vis0}} \mid s_{\text{env}}\right) p\left(s_{\text{env}}\right) ds_{\text{env}}\right) \end{aligned} \tag{19}$$

where the integral with respect to $s_{\text{env}}$ characterizes the contribution of visual signals to the inference of self-motion. A key point here is that $x_{\text{vis}}$ depends both on $s_{\text{self}}$ and $s_{\text{env}}$, while $x_{\text{vis0}}$ depends only on $s_{\text{env}}$. Hence, the observer can infer $s_{\text{env}}$ from $x_{\text{vis0}}$, which can be then used to isolate $s_{\text{self}}$ from $x_{\text{vis}}$. The heading estimate given that environmental motion is constant can be then calculated as:

$$\hat{s}_{\text{self},C=1} = \int s_{\text{self}}\, p\left(x_{\text{vest}} \mid s_{\text{self}}\right)\left(\int p\left(x_{\text{vis}} \mid s_{\text{self}}, s_{\text{env}}\right) p\left(x_{\text{vis0}} \mid s_{\text{env}}\right) p\left(s_{\text{env}}\right) ds_{\text{env}}\right) p\left(s_{\text{self}}\right) ds_{\text{self}} \tag{20}$$

On the other hand, when the environmental motion has changed, only $x_{\text{vest}}$ and $x_{\text{vis}}$ provide relevant information about the observer's heading, so we can safely omit $x_{\text{vis0}}$. Thus, the likelihood can be rewritten as:

$$\begin{aligned} p\left(x_{\text{vest}}, x_{\text{vis}}, x_{\text{vis0}} \mid s_{\text{self}}, C=2\right) &= \int p\left(x_{\text{vest}}, x_{\text{vis}}, x_{\text{vis0}} \mid s_{\text{self}}, s_{\text{env}}\right) p\left(s_{\text{env}}\right) ds_{\text{env}} \\ &= \int p\left(x_{\text{vest}} \mid s_{\text{self}}\right) p\left(x_{\text{vis}} \mid s_{\text{self}}, s_{\text{env}}\right) p\left(s_{\text{env}}\right) ds_{\text{env}} \\ &= p\left(x_{\text{vest}} \mid s_{\text{self}}\right)\left(\int p\left(x_{\text{vis}} \mid s_{\text{self}}, s_{\text{env}}\right) p\left(s_{\text{env}}\right) ds_{\text{env}}\right) \end{aligned} \tag{21}$$

where the integral with respect to $s_{\text{env}}$ characterizes the contribution of visual signals to the inference of self-motion. This time, $x_{\text{vis}}$ still depends on both $s_{\text{self}}$ and $s_{\text{env}}$, but there is no other sensory information about $s_{\text{env}}$, leaving the observer to rely solely on the prior $p\left(s_{\text{env}}\right)$ to infer $s_{\text{env}}$ in order to isolate $s_{\text{self}}$ from $x_{\text{vis}}$. The heading estimate given that environmental motion is independent before and during self-motion can be then calculated as:

$$\hat{s}_{\text{self},C=2} = \int s_{\text{self}}\, p\left(x_{\text{vest}}|s_{\text{self}}\right) \left(\int p\left(x_{\text{vis}}|s_{\text{self}}, s_{\text{env}}\right) p\left(s_{\text{env}}\right) ds_{\text{env}}\right) p\left(s_{\text{self}}\right) ds_{\text{self}} \tag{22}$$

All terms in Eqs 20 and 22 are Gaussians, allowing for analytic solutions, as shown in Eq 2.

## Integration (Int) model

We consider an alternative model in which the observer always assumes the environmental motion to be constant before and during self-motion. Therefore, visual signals collected before and during self-motion are mandatorily integrated [109] by subtracting the posterior estimate of environmental motion before self-motion from the visual signal collected during self-motion. This is a special case of the Contextual Causal Inference model in which the prior probability that the environmental motion would be constant, $p_{\text{constant}}$, is set to 1. Thus,

$$\hat{s}_{\text{self}} = \frac{\frac{x_{\text{vest}}}{\sigma_{\text{vest}}^2} + \frac{\mu_K}{\sigma_K^2} + \frac{0}{\sigma_{\text{self}}^2}}{\frac{1}{\sigma_{\text{vest}}^2} + \frac{1}{\sigma_K^2} + \frac{1}{\sigma_{\text{self}}^2}} \tag{23}$$

where

$$\mu_K = x_{\text{vis}} - \frac{\sigma_{\text{env}}^2}{\sigma_{\text{env}}^2 + \sigma_{\text{vis0}}^2} x_{\text{vis0}} \tag{24}$$

$$\sigma_K^2 = \sigma_{\text{vis}}^2 + \frac{\sigma_{\text{env}}^2 \sigma_{\text{vis0}}^2}{\sigma_{\text{env}}^2 + \sigma_{\text{vis0}}^2} \tag{25}$$

## Segregation (Seg) model

In this model, the observer always assumes the environmental motion to be independent before and during self-motion. As a result, the observer completely disregards the visual signal collected before self-motion and interprets the visual signal collected during self-motion as being entirely generated by self-motion, albeit with increased uncertainty. This is a special case of the Contextual Causal Inference model in which the prior probability that the environmental motion would be constant, $p_{\text{constant}}$, is set to 0. Thus,

$$\hat{s}_{\text{self}} = \frac{\frac{x_{\text{vest}}}{\sigma_{\text{vest}}^2} + \frac{\mu_K}{\sigma_K^2} + \frac{0}{\sigma_{\text{self}}^2}}{\frac{1}{\sigma_{\text{vest}}^2} + \frac{1}{\sigma_K^2} + \frac{1}{\sigma_{\text{self}}^2}} \tag{26}$$

where

$$\mu_K = x_{\text{vis}} \tag{27}$$

$$\sigma_K^2 = \sigma_{\text{vis}}^2 + \sigma_{\text{env}}^2 \tag{28}$$

## Covariance (Cov) model

In this model, the observer assumes that environmental motion before and during self-motion covary with each other [30–34]. Specifically, environmental motion before and during self-motion, $\begin{bmatrix} s_{\text{env0}} \\ s_{\text{env}} \end{bmatrix}$, is assumed to follow a bivariate zero-mean

Gaussian prior distribution $\mathcal{N}\left(\begin{bmatrix} 0 \\ 0 \end{bmatrix}, \begin{bmatrix} \sigma_{env}^2 & \rho\sigma_{env}^2 \\ \rho\sigma_{env}^2 & \sigma_{env}^2 \end{bmatrix}\right)$ where $\rho$ is a Pearson correlation coefficient. We treat $\rho$ as a free parameter, as it has been shown to depend on the statistics of the task but with a significant bias [32]. The Integration model and Segregation model are a special case of the Covariance model in which the correlation, $\rho$, is set to 0 and 1, respectively.

As in other models, the observer computes the optimal heading estimate that minimizes the posterior expected loss:

$$\hat{s}_{self} = \underset{\hat{s}_{self}}{\text{argmin}} \int \ell\left(\hat{s}_{self}, s_{self}\right) p\left(s_{self}|x_{vest}, x_{vis}, x_{vis0}\right) ds_{self} \tag{29}$$

where we assume a squared error loss. The posterior of self-motion is proportional to the product of the likelihood multiplied by the prior.

$$p\left(s_{self}|x_{vest}, x_{vis}, x_{vis0}\right) \propto p\left(x_{vest}, x_{vis}, x_{vis0}|s_{self}\right) p\left(s_{self}\right) \tag{30}$$

All available sensory signals provide relevant information about self-motion, and $s_{env0}$ and $s_{env}$ are distinct entities but not statistically independent. Thus, the likelihood can be rewritten as:

$$\begin{aligned} p\left(x_{vest}, x_{vis}, x_{vis0}|s_{self}\right) &= \iint p\left(x_{vest}, x_{vis}, x_{vis0}|s_{self}, s_{env}, s_{env0}\right) p\left(s_{env}, s_{env0}\right) ds_{env} ds_{env0} \\ &= \iint p\left(x_{vest}|s_{self}\right) p\left(x_{vis}|s_{self}, s_{env}\right) p\left(x_{vis0}|s_{env0}\right) p\left(s_{env0}|s_{env}\right) p\left(s_{env}\right) ds_{env} ds_{env0} \\ &= p\left(x_{vest}|s_{self}\right) \left( \int p\left(x_{vis}|s_{self}, s_{env}\right) \left( \int p\left(x_{vis0}|s_{env0}\right) p\left(s_{env0}|s_{env}\right) ds_{env0} \right) p\left(s_{env}\right) ds_{env} \right) \end{aligned} \tag{31}$$

where the integral with respect to $s_{env0}$ characterizes the contribution of visual signal collected before self-motion to the inference of environmental motion during self-motion, and the integral with respect to $s_{env}$ characterizes the joint contribution of visual signals to the inference of self-motion. A key point here is that $x_{vis}$ depends on both $s_{self}$ and $s_{env}$, and $s_{env}$ is correlated with $s_{env0}$. Hence, the observer can use a conditional prior to infer $s_{env}$ from $x_{vis0}$, which in turn, combined with $x_{vis}$, contributes to the inference of $s_{self}$. All terms in Eq 31 are Gaussian, as well as the prior $p\left(s_{self}\right)$ in Eq 30, allowing for an analytic solution:

$$\hat{s}_{self} = \frac{\frac{x_{vest}}{\sigma_{vest}^2} + \frac{\mu_K}{\sigma_K^2} + \frac{0}{\sigma_{self}^2}}{\frac{1}{\sigma_{vest}^2} + \frac{1}{\sigma_K^2} + \frac{1}{\sigma_{self}^2}} \tag{32}$$

where

$$\mu_K = x_{vis} - \frac{\rho\sigma_{env}^2}{\sigma_{env}^2 + \sigma_{vis0}^2} x_{vis0} \tag{33}$$

$$\sigma_K^2 = \sigma_{vis}^2 + \sigma_{env}^2 \left( 1 - \frac{\rho^2 \sigma_{env}^2}{\sigma_{env}^2 + \sigma_{vis0}^2} \right) \tag{34}$$

**Fixed weight (Fix) model**

In this model, the observer integrates sensory signals using fixed weights:

$$\hat{s}_{self} = \alpha_{vest} x_{vest} + \alpha_{mom} x_{vis} + \alpha_{cont} \left( x_{vis} - x_{vis0} \right) \tag{35}$$

where sensory signals, $x_{\text{vest}}$, $x_{\text{vis}}$ and $x_{\text{vis0}}$, are assumed to be generated in the same way as in other models. Since this is rather a descriptive model with the weights, $\alpha_{\text{vest}}$, $\alpha_{\text{mom}}$ and $\alpha_{\text{cont}}$, no longer based on the generative model, this model does not consider the priors on environmental and self-motion.

### Heuristic model

In this model, the observer performs non-Bayesian causal inference and does not incorporate uncertainty in the prior or sensory information. Instead, the observer relies on simple heuristics to decide whether environmental motion has remained constant before and during self-motion. Specifically, the observer compares the absolute difference between the vestibular signal and the visual estimate of heading direction assuming each causal structure, and devotes entirely to a single heading estimate associated with the winning one:

$$\hat{s}_{\text{self}} = \begin{cases} \hat{s}_{\text{self},C=1}, & \left|x_{\text{vest}} - \mu_{K,C=1}\right| \geq \left|x_{\text{vest}} - \mu_{K,C=2}\right| \\ \hat{s}_{\text{self},C=2}, & \left|x_{\text{vest}} - \mu_{K,C=1}\right| < \left|x_{\text{vest}} - \mu_{K,C=2}\right| \end{cases} \tag{36}$$

where $\hat{s}_{\text{self},C}$ and $\mu_K$ are given in Eqs 2 and 3, respectively.

### Winner-take-all model

In this model, the observer computes the posterior probability that environmental motion has remained constant, but instead of combining the two heading estimates weighted by the posterior probabilities, the observer devotes entirely to a single heading estimate associated with a posteriori more probable causal structure (i.e., model selection):

$$\hat{s}_{\text{self}} = \begin{cases} \hat{s}_{\text{self},C=1}, & p\left(C = 1|x_{\text{vest}}, x_{\text{vis}}, x_{\text{vis0}}\right) \geq p\left(C = 2|x_{\text{vest}}, x_{\text{vis}}, x_{\text{vis0}}\right) \\ \hat{s}_{\text{self},C=2}, & p\left(C = 1|x_{\text{vest}}, x_{\text{vis}}, x_{\text{vis0}}\right) < p\left(C = 2|x_{\text{vest}}, x_{\text{vis}}, x_{\text{vis0}}\right) \end{cases} \tag{37}$$

where $\hat{s}_{\text{self},C}$ and $p\left(C|x_{\text{vest}}, x_{\text{vis}}, x_{\text{vis0}}\right)$ are given in Eqs 2 and 10, respectively.

### Momentary causal inference (MCI) model

Lastly, we consider a conventional causal inference model of multisensory heading perception in literature [16–18] that does not take into account visual signal collected before self-motion and simply infers whether visual and vestibular signals collected during self-motion originate from the same (i.e., $C = 1$) or different (i.e., $C = 2$) cause. Note that derivations of this model have been described before [16–18,25], and we only include them here to make the paper self-contained.

In this model, both $s_{\text{self}}$ and $s_{\text{env}}$ are assumed to follow a zero-mean Gaussian distribution $\mathcal{N}\left(0, \sigma_{\text{self}}^2\right)$. A noisy vestibular signal is then drawn from a Gaussian measurement distribution $\mathcal{N}\left(s_{\text{self}}, \sigma_{\text{vest}}^2\right)$, and a noisy visual signal is drawn from a Gaussian measurement distribution $\mathcal{N}\left(s_{\text{env}}, \sigma_{\text{vis}}^2\right)$ with $\sigma_{\text{vis}}^2 = w_{\text{vis}}^2 \left|s_{\text{env}}\right|^2$. Note that here $s_{\text{env}}$ does not necessarily represent environmental motion but instead represents an arbitrary state of the world that gives rise to a visual signal. The causal structure, $C \in \{1, 2\}$, determines whether $s_{\text{self}}$ and $s_{\text{env}}$ are from one cause (i.e., $C = 1$) or two causes (i.e., $C = 2$) by drawing from a binomial distribution with $p(C = 1) = p_{\text{common}}$. If there is one cause (i.e., $C = 1$), $s_{\text{self}}$ and $s_{\text{env}}$ take the same value, and if there are two causes (i.e., $C = 2$), $s_{\text{self}}$ and $s_{\text{env}}$ are independent.

Following the same logic in the Contextual Causal Inference model, an optimal heading estimate is given by:

$$\hat{s}_{\text{self}} = p\left(C = 1|x_{\text{vest}}, x_{\text{vis}}\right) \hat{s}_{\text{self},C=1} + p\left(C = 2|x_{\text{vest}}, x_{\text{vis}}\right) \hat{s}_{\text{self},C=2} \tag{38}$$

Applying Bayes' rule, for $p\left(C = 1|x_{\text{vest}}, x_{\text{vis}}\right)$, we obtain:

$$p\left(C = 1|x_{\text{vest}}, x_{\text{vis}}\right) = \frac{p(x_{\text{vest}}, x_{\text{vis}}|C=1)p_{\text{common}}}{p(x_{\text{vest}}, x_{\text{vis}}|C=1)p_{\text{common}} + p(x_{\text{vest}}, x_{\text{vis}}|C=2)(1-p_{\text{common}})} \tag{39}$$

Analogously, for $p\left(C=2|x_{\text{vest}}, x_{\text{vis}}\right)$, we obtain:

$$p\left(C=2|x_{\text{vest}}, x_{\text{vis}}\right) = \frac{p(x_{\text{vest}}, x_{\text{vis}}|C=2)(1-p_{\text{common}})}{p(x_{\text{vest}}, x_{\text{vis}}|C=1)p_{\text{common}}+p(x_{\text{vest}}, x_{\text{vis}}|C=2)(1-p_{\text{common}})} \tag{40}$$

When $C=1$, we substitute $s_{\text{env}}$ with $s_{\text{self}}$, and thus the likelihood can be rewritten as:

$$p\left(x_{\text{vest}}, x_{\text{vis}}|C=1\right) = \int p\left(x_{\text{vest}}|s_{\text{self}}\right) p\left(x_{\text{vis}}|s_{\text{self}}\right) p\left(s_{\text{self}}\right) ds_{\text{self}} \tag{41}$$

All terms in the integral are Gaussian, so we can write down an analytic solution:

$$p\left(x_{\text{vest}}, x_{\text{vis}}|C=1\right) = \frac{1}{2\pi\sqrt{\sigma_{\text{vest}}^2\sigma_{\text{vis}}^2+\sigma_{\text{vest}}^2\sigma_{\text{self}}^2+\sigma_{\text{vis}}^2\sigma_{\text{self}}^2}} \exp\left(-\frac{1}{2}\frac{(x_{\text{vest}}-x_{\text{vis}})^2\sigma_{\text{self}}^2+x_{\text{vest}}^2\sigma_{\text{vis}}^2+x_{\text{vis}}^2\sigma_{\text{vest}}^2}{\sigma_{\text{vest}}^2\sigma_{\text{vis}}^2+\sigma_{\text{vest}}^2\sigma_{\text{self}}^2+\sigma_{\text{vis}}^2\sigma_{\text{self}}^2}\right) \tag{42}$$

When $C=2$, $x_{\text{vest}}$ and $x_{\text{vis}}$ are independent of each other, and we thus obtain a product of two factors:

$$p\left(x_{\text{vest}}, x_{\text{vis}}|C=2\right) = \iint p\left(x_{\text{vest}}|s_{\text{self}}\right) p\left(x_{\text{vis}}|s_{\text{env}}\right) p\left(s_{\text{self}}\right) p\left(s_{\text{env}}\right) ds_{\text{self}}ds_{\text{env}}$$

$$= \left(\int p\left(x_{\text{vest}}|s_{\text{self}}\right) p\left(s_{\text{self}}\right) ds_{\text{self}}\right) \left(\int p\left(x_{\text{vis}}|s_{\text{env}}\right) p\left(s_{\text{env}}\right) ds_{\text{env}}\right) \tag{43}$$

Again, all terms in the integral are Gaussian, so we can write down an analytic solution:

$$p\left(x_{\text{vest}}, x_{\text{vis}}|C=2\right) = \frac{1}{2\pi\sqrt{(\sigma_{\text{vest}}^2+\sigma_{\text{self}}^2)(\sigma_{\text{vis}}^2+\sigma_{\text{self}}^2)}} \exp\left(-\frac{1}{2}\left(\frac{x_{\text{vest}}^2}{\sigma_{\text{vest}}^2+\sigma_{\text{self}}^2} + \frac{x_{\text{vis}}^2}{\sigma_{\text{vis}}^2+\sigma_{\text{self}}^2}\right)\right) \tag{44}$$

Now we compute the heading estimates given each causal structure. When the visual and vestibular signals are from the same cause, both $x_{\text{vest}}$ and $x_{\text{vis}}$ provide relevant information about the observer's heading. Therefore,

$$\hat{s}_{\text{self},C=1} = \int s_{\text{self}}\, p\left(x_{\text{vest}}|s_{\text{self}}\right) p\left(x_{\text{vis}}|s_{\text{self}}\right) p\left(s_{\text{self}}\right) ds_{\text{self}} = \frac{\frac{x_{\text{vest}}}{\sigma_{\text{vest}}^2}+\frac{x_{\text{vis}}}{\sigma_{\text{vis}}^2}+\frac{0}{\sigma_{\text{self}}^2}}{\frac{1}{\sigma_{\text{vest}}^2}+\frac{1}{\sigma_{\text{vis}}^2}+\frac{1}{\sigma_{\text{self}}^2}} \tag{45}$$

On the other hand, when $x_{\text{vest}}$ and $x_{\text{vis}}$ are independent, only $x_{\text{vest}}$ provide relevant information about the observer's heading. Therefore,

$$\hat{s}_{\text{self},C=2} = \int s_{\text{self}}\, p\left(x_{\text{vest}}|s_{\text{self}}\right) p\left(s_{\text{self}}\right) ds_{\text{self}} = \frac{\frac{x_{\text{vest}}}{\sigma_{\text{vest}}^2}+\frac{0}{\sigma_{\text{self}}^2}}{\frac{1}{\sigma_{\text{vest}}^2}+\frac{1}{\sigma_{\text{self}}^2}} \tag{46}$$

### Model fitting

All of the above models specify a deterministic mapping from sensory measurements $x_{\text{vest}}$, $x_{\text{vis}}$ and/or $x_{\text{vis0}}$ to a heading estimate $\hat{s}_{\text{self}}$. Since the observers' internal sensory measurements are psychophysically unobservable, we eliminated the dependence on these variables by integrating over the hidden variables (i.e., marginalization).

$$p\left(\hat{s}_{\text{self}}|s_{\text{self}}, s_{\text{env}}, s_{\text{env0}}\right) = \iiint p\left(\hat{s}_{\text{self}}|x_{\text{vest}}, x_{\text{vis}}, x_{\text{vis0}}\right) p\left(x_{\text{vest}}|s_{\text{self}}\right) p\left(x_{\text{vis}}|s_{\text{self}}, s_{\text{env}}\right) p\left(x_{\text{vis0}}|s_{\text{env0}}\right) dx_{\text{vest}}dx_{\text{vis}}dx_{\text{vis0}} \tag{47}$$

We assumed that observers' heading estimates were independent across trials and thus expressed the joint log likelihood of heading estimates across all trials as the sum of the individual log likelihoods:

$$\log p\left(\{\hat{s}_{\text{self}}^{(i,j)}\}|\{s_{\text{self}}^{(i)}\}, \{s_{\text{env}}^{(i)}\}, \{s_{\text{env0}}^{(i)}\}, \theta\right) = \sum_i \sum_j \log p\left(\hat{s}_{\text{self}}^{(i,j)}|s_{\text{self}}^{(i)}, s_{\text{env}}^{(i)}, s_{\text{env0}}^{(i)}, \theta\right)$$

(48)

where the subscripts $i$ and $j$ are indices for stimulus conditions and trials within each stimulus condition, respectively, and $\theta$ is model parameters. Contextual Causal Inference model and Winner-Take-All model had five free parameters, $\theta = \{\sigma_{\text{self}}, \sigma_{\text{env}}, \sigma_{\text{vest}}, w_{\text{vis}}, p_{\text{constant}}\}$, Integration model, Segregation model and Heuristic model had four, $\theta = \{\sigma_{\text{self}}, \sigma_{\text{env}}, \sigma_{\text{vest}}, w_{\text{vis}}\}$, Covariance model had five, $\theta = \{\sigma_{\text{self}}, \sigma_{\text{env}}, \sigma_{\text{vest}}, w_{\text{vis}}, \rho\}$, Fixed Weight model had five, $\theta = \{\sigma_{\text{vest}}, w_{\text{vis}}, \alpha_{\text{vest}}, \alpha_{\text{mom}}, \alpha_{\text{cont}}\}$, and Momentary Causal Inference model had four, $\theta = \{\sigma_{\text{self}}, \sigma_{\text{vest}}, w_{\text{vis}}, p_{\text{common}}\}$. Unlike the CCI model, parameter estimates for some alternative models were unstable for the most observers. Therefore, we further constrained the model fitting with priors on the model parameters:

$$\log p\left(\theta|\{s_{\text{self}}^{(i)}\}, \{s_{\text{env}}^{(i)}\}, \{s_{\text{env0}}^{(i)}\}, \{\hat{s}_{\text{self}}^{(i,j)}\}\right) \propto \log p\left(\{\hat{s}_{\text{self}}^{(i,j)}\}|\{s_{\text{self}}^{(i)}\}, \{s_{\text{env}}^{(i)}\}, \{s_{\text{env0}}^{(i)}\}, \theta\right) + \log p(\theta)$$

(49)

Specifically, we constrained $\sigma_{\text{self}}, \sigma_{\text{env}}, \sigma_{\text{vest}}$ and $w_{\text{vis}}$ with independent log-normal priors whose means on the log scale were set to 1, 1, 1 and −2, respectively, and standard deviations to 1, allocating 95% of prior mass roughly between 1/7 and 7 times the median. All the other parameters that have both upper and lower bounds by definition were given uninformative flat priors.

We solved the integrals in Eq 47 via a Monte Carlo sampling, drawing 1,000 samples of $x_{\text{vest}}$, $x_{\text{vis}}$ and $x_{\text{vis0}}$ from the measurement distributions for each stimulus condition, $s_{\text{self}}^{(i)}$, $s_{\text{env}}^{(i)}$ and $s_{\text{env0}}^{(i)}$. The log likelihood of five heading estimates for each stimulus condition was then computed from the Monte Carlo samples via a kernel density estimation. We used Eq 49 to minimize the negative log posterior probability of model parameters given all heading estimates measured psychophysically for each observer. Since our objective function was inherently stochastic due to the Monte Carlo sampling, we used Bayesian adaptive direct search [110] to find model parameters that minimize the expected value of the stochastic objective function, smoothing the observed function values via a Gaussian process. After the optimization, we calculated the log likelihood by averaging 100 evaluations of the objective function and subtracting the log prior probability. We evaluated the success of the fitting exercise by repeating the search with different initial values and confirmed that the fitting procedure was stable with respect to initial values. Parameter estimates are summarized in Tables A and B in S1 Supporting information.

For the simulation of model observers' behavior shown in Figs 3B and 4C (and analysis on the simulated behavior shown in Figs 3C, 4B, 4D and 5B), we used the same trials that the human observers went through in the experiment. For the model prediction shown in Figs 2 and B in S1 Supporting information, we simulated 10,000 heading estimates for each observer and each unique stimulus condition and took the average. The coefficient of determination ($R^2$) for each observer was computed using this model prediction.

### Variable weight analysis

To determine whether human observers performed linear or nonlinear sensory integration, we tested if the weights on $\hat{s}_{\text{self},C=1}$ and $\hat{s}_{\text{self},C=2}$ vary depending on the true causal structure, $C$. Instead of computing the posterior probability of the causal structure (Eq 10), we used one of two weight parameters, $\alpha_1$ and $\alpha_2$, depending on whether environmental motion before and during self-motion was actually constant:

$$\hat{s}_{\text{self}} = \begin{cases} \alpha_1 \hat{s}_{\text{self},C=1} + (1-\alpha_1) \hat{s}_{\text{self},C=2}, & s_{\text{env0}} = s_{\text{env}} \\ \alpha_2 \hat{s}_{\text{self},C=1} + (1-\alpha_2) \hat{s}_{\text{self},C=2}, & s_{\text{env0}} \neq s_{\text{env}} \end{cases}$$

(50)

where $\hat{s}_{\text{self},C}$ is given in Eq 2. Note that this is not an observer model, as human observers do not have direct access to the true state of the world, $s_{\text{env0}}$ or $s_{\text{env}}$. The purpose of this analysis was not to build another observer model, but to reveal

a model-based diagnostic pattern in the data. We fit six free parameters, $\theta = \{\sigma_{self}, \sigma_{env}, \sigma_{vest}, w_{vis}, \alpha_1, \alpha_2\}$, separately to human and model observers' heading estimates, using the same fitting methods described above.

## Supporting information

**S1 Supporting Information. Supplemental methods, figures and tables.**
(PDF)

**S1 Movie. Example visual motion stimulus (size: 60 × 60°; speed: 0°/s).**
(MOV)

**S2 Movie. Example visual motion stimulus (size: 60 × 60°; speed: 5°/s).**
(MOV)

**S3 Movie. Example visual motion stimulus (size: 60 × 60°; speed: 10°/s).**
(MOV)

## Author contributions

**Conceptualization:** Liana Nafisa Saftari, Jongmin Moon, Oh-Sang Kwon.

**Formal analysis:** Liana Nafisa Saftari, Jongmin Moon.

**Funding acquisition:** Oh-Sang Kwon.

**Investigation:** Liana Nafisa Saftari.

**Methodology:** Jongmin Moon.

**Supervision:** Oh-Sang Kwon.

**Visualization:** Liana Nafisa Saftari, Jongmin Moon.

**Writing – original draft:** Liana Nafisa Saftari, Jongmin Moon.

**Writing – review & editing:** Liana Nafisa Saftari, Jongmin Moon, Oh-Sang Kwon.

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
