## [Decision Letter · Decision Letter 0]

21 Apr 2025

Environmental motion presented ahead of self-motion modulates heading direction estimation

PLOS Computational Biology

Dear Dr. Kwon,

Thank you for submitting your manuscript to PLOS Computational Biology. After careful consideration, we feel that it has merit but does not fully meet PLOS Computational Biology's publication criteria as it currently stands. Therefore, we invite you to submit a revised version of the manuscript that addresses the points raised during the review process.

Please submit your revised manuscript within 60 days Jun 21 2025 11:59PM. If you will need more time than this to complete your revisions, please reply to this message or contact the journal office at ploscompbiol@plos.org. Please include the following items when submitting your revised manuscript:

We look forward to receiving your revised manuscript.

Kind regards,

Paul Bays

Academic Editor

PLOS Computational Biology

Lyle Graham

Section Editor

PLOS Computational Biology

**Additional Editor Comments :**

As you will see, the Reviewers have positive things to say about your manuscript but also raised a number of concerns that will need to be fully and carefully addressed before a final decision can be made.

**Journal Requirements:**

At this stage, the following Authors/Authors require contributions: Liana Nafisa Saftari, Jongmin Moon, and Oh-Sang Kwon. Please ensure that the full contributions of each author are acknowledged in the "Add/Edit/Remove Authors" section of our submission form.

4) Your manuscript is missing the following section heading: Abstract.  Please ensure all required sections are present and in the correct order. Make sure section heading levels are clearly indicated in the manuscript text, and limit sub-sections to 3 heading levels. An outline of the required sections can be consulted in our submission guidelines here:

5) Please upload all main figures as separate Figure files in .tif or .eps format. For more information about how to convert and format your figure files please see our guidelines: 

Potential Copyright Issues:

i) Figures 1A, and 1B. Please confirm whether you drew the images / clip-art within the figure panels by hand. If you did not draw the images, please provide (a) a link to the source of the images or icons and their license / terms of use; or (b) written permission from the copyright holder to publish the images or icons under our CC BY 4.0 license. Alternatively, you may replace the images with open source alternatives. See these open source resources you may use to replace images / clip-art:

7) Please amend your detailed Financial Disclosure statement. This is published with the article. It must therefore be completed in full sentences and contain the exact wording you wish to be published.

2) State what role the funders took in the study. If the funders had no role in your study, please state: "The funders had no role in study design, data collection and analysis, decision to publish, or preparation of the manuscript.".

**Reviewers' comments:**

Reviewer's Responses to Questions

Reviewer #1: This paper examines how visual motion before a self-motion influences heading perception. Building on past work that typically focuses only on concurrent visual–vestibular inputs, the authors present a paradigm where participants are moved on a platform while viewing visual stimuli that vary in speed and velocity profile prior to the movement. Their main finding is that these pre-movement stimuli bias subsequent heading estimates. They propose a Bayesian causal-inference model that combines (1) vestibular information, (2) concurrent visual motion, and (3) contextual (pre-movement) visual signals. Their conclusion is that the brain integrates cues about self-motion across time.

Main criticisms;

1. Inadequate presentation of individual-participant data

Group-level Averaging vs. Individual Patterns. The paper relies heavily on group-averaged results (e.g., Figure 2), but the authors’ claims emphasize that heading perception and causal inference are implemented in individual brains. If the claims are indeed about individuals, more subject-level detail is essential to confirm that effects hold systematically across participants (e.g., violin plots or per-subject fits).

Model Fits and Variability. The manuscript would benefit from showing how well the Bayesian model explains each individual’s data (variance explained, parameter values, etc.). The current group-level approach may mask considerable individual differences, and it raises concerns about underpowered inferences if the main statistical tests are done at the group level.

2. Overprocessed data

The data presented in Fig 2. appears to have averaged over the 10 different motion directions that were presented. Why are biases not shown as a function of that direction?

3. Clarity on “Optimal” Estimates and the Underlying Loss Function

Lines 114 and 120: Confusion About “Optimal.” The text uses “optimal estimate” in contexts that invoke both mean-based (L2 loss) and MAP (L0 or similar) solutions, but these concepts have distinct mathematical meanings. For instance, line 114 refers to a model-averaging approach (optimal for L2 loss), yet line 120 seems to imply a MAP approach (optimal under a different loss function).

Line 134: “Optimal Solution That Maximizes Posterior Probability.” Here again, the authors equate maximizing the posterior with an “optimal solution,” but this is only correct for L0 loss. While these in accuracies don’t appear consequential for this particular manuscript since L0, L1, and L2 all imply the same for the Gaussian posteriors assumed here, the manuscript’s terminology should be made more precise.

4. Bayesian Model vs. Regression Models: Significance of Coefficients

3a) The paper appears to use two different kinds of models, a regression model, and a Bayesian model. While the equations are provided for the latter, the equations for the former would be helpful.

3b) I assume the Bayesian model was fit to individual observers but virtually no details were given on the results of this fitting: how many parameters were fit, what were their values, how good was the fit on an individual subject level, etc

3c) The manuscript reports statistical significance (“stars”) for regression coefficients associated with data but it’s unclear whether they apply to the model fits (Fig 2b).

3d) Relatedly, the authors discuss linear regression coefficients for different “strategies” (e.g., relying purely on vestibular information vs. adding contextual vision), but it is unclear how these regression values map onto the actual Bayesian model parameters.

5. Evidence for Causal Inference vs. Forced Integration

Although the authors frame the Bayesian model as implementing causal inference (i.e., deciding whether environmental motion remains constant or changes over time), it is not entirely clear that there is strong behavioral evidence for a true “inference” process as opposed to simpler forced-integration strategies. Additional model comparisons would be valuable. Demonstrating that a forced-integration or fixed-weights model clearly performs worse than their “causal” approach would strengthen the central causal-inference claim. Similarly, what qualitative signatures would be expected from these alternative models in terms of the data obtained from this experiment?

6. Undiscussed Parameters and Fits

How many free parameters does the Bayesian model have, and how are they estimated? The paper mentions a Monte Carlo procedure but leaves open questions about prior assumptions, parameter fitting details (e.g., prior variance, Weber fraction, probability of the environment being static), and how the authors handled the stochastic likelihood of each trial (lines 351–353). An expanded methods section would help solidify confidence in the modeling approach.

7. Limited Discussion of Related Literature

The paper would benefit from an expanded discussion of related works in particular in the motion domain (“flow parsing” by Warren & Rushton & colleagues) and the works that study the “temporal binding window”, i.e. the temporal discrepancy allowed before visual and temporal signals aren’t combined - closely related to “temporal context” in this paper.

A brief but explicit explanation of how the likelihood was computed—particularly how the “stochastic likelihood” is integrated across trials—would add clarity.

Reviewer #2: The authors have asked an important question here and have done an interesting experiment to address it. Most studies of cue integration have only considered how the brain combines two sources of information that are presented simultaneously. But, in real life, the onset and offset of different sources of information may not be synchronous, and there may be valuable information available in how a cue gets integrated with others depending on how it was available previously. In this case, the authors examine perception of heading in response to visual and vestibular cues, and the examine how a previously available visual cue influences perception during a period of time in which both visual and vestibular cues are available.

This is a valuable topic to investigate, and there are many things to like about the study. However, it also has some important limitations that need to be investigated and discussed further. There are also some crucial technical details missing that are important to evaluate the quality of the stimuli.

Major issues:

1) It is not clear how the authors matched the dynamics and latency of the visual and vestibular stimuli. They show both as having a roughly trapezoidal waveform that is synchronous. But any motion platform is going to have dynamics and some latency. Those dynamics will make the actual movement trajectory deviate from the command signal that is sent to move the platform. So if the authors are equating the platform command signal with their visual motion trajectory, then the end result is going to be visual and vestibular stimuli that are not actually the same. It is crucial to get this right, otherwise subjects might be able to tell that the visual motion they are seeing is not consistent with the platform motion they are feeling, which would compromise the design of the study. The authors need to explain how they addressed these issues and provide evidence that the two types of stimulation indeed have the same timing and the same temporal profile.

2) The stimulus situation that the authors have concocted is naturalistic in some ways but unnatural in others. Specifically, all of the visual motion that subjects see in their visual field is unidirectional and caused by environmental motion. Apart from a situation in which a moving object is so large and so nearby that it takes up the entire visual field, this is something that never happens in real life. In real life, the visual motion field includes some components that associated with self-motion (i.e., stationary background elements) and some that reflect both self-motion and object (environmental) motion (e.g., the example that the authors themselves give in the first paragraph of the Introduction). In those situations, there are other mechanisms, such as optic flow parsing, that also contribute to how we perceive object motion and self-motion. The probabilities that subjects apply to the different motion inputs in this experiment might therefore be shaped by this design, or their actual priors (formed from natural experience) might not be very compatible with the stimulus situation that is depicted. I’m not suggesting that the authors need to re-do their experiments using a more naturalistic stimulus, but I do think that it is necessary that they discuss these limitations at some length and provide some appropriate caveats related to how their findings might generalize.

3) The authors fit the behavioral data with a causal inference model. In this model, the two causal interpretations are that environmental motion is constant before and during self-motion (C=1) or that the environmental motion changes in the time period before vs. during self-motion (C=2). There are a few aspects of this that are awkward. A) It is a bit strange to use this type of model when the subjects are never asked to make this causal judgment. Hence, the data may not really constrain fits of this type of model very well. This is a major limitation that needs to be at least discussed explicitly. B) The authors assume that the subjects have a prior probability on C=1 vs C=2 that is constant and matched to the stimulus statistics. But there is a good chance that this might not be the case. Previous studies have shown that different observers can have quite different prior probabilities over differ possible motion causes (e.g., Shivkumar et al. 2024). So this is a weakness that needs to be called out. C) The authors’ prediction under the “momentary strategy” seems to assume that the environment is moving before self-motion and that it abruptly becomes stationary during self-motion. In reality, it seems possible that the subjects perceive the background to be moving in the world at one speed before self-motion and at another speed during self-motion. This percept could be fit by the authors model, but it doesn’t mean that it is a good model for what subjects experience. So this should be discussed. And if there is any temporal mismatch in the visual and vestibular stimuli, that would greatly complicate the interpretation of this. It would have been useful for the authors to at least debrief their subjects about their interpretation of the stimuli.

4) Just because the predictions of a Bayesian model fit the data doesn’t demonstrate that subjects are performing a Bayesian computation. In this study, since the authors did not vary the reliability of the different motion cues, it is not clear to me that one could reject other simpler models. Given that the data and prediction curves in Fig. 2A are just linear functions of speed, it seems like other models that do a weighted sum of the cues might also fit the data well. Can the authors demonstrate that a Bayesian model is necessary in this context? Can they demonstrate that it is possible to rule out simpler models?

Other specific issues:

5) Line 172: “The causal inference framework has been wildly employed…” I presume the authors meant “widely employed”?

6) Given that the visual stimulus is not a typical type of optic flow field, it is important that the authors include a video that shows examples of the stimuli.

7) Lines 273-275: This part about symmetry was hard to follow. Please rewrite and clarify.

8) Lines 278-282: The explanation of the momentary and contextual strategies, both here and in the main text, was not very clearly explained. I had to read these sections several times to get it. Perhaps a diagram that illustrates these strategies would aid some text revisions. Or it may be useful to give equations to make the predictions of these strategies more explicit.

9) The authors’ model includes a zero-mean Gaussian prior on environmental motion. While such a slow speed prior has been used by others, recent studies (e.g., Shivkumar et al. 2024; Penaloza et al. 2024) have argued that this is not a good prior for object motion. Most objects in a scene are stationary and have exactly zero speed. Those that are moving may be drawn from a zero-mean Gaussian prior. Thus, others have argued for the use of a bipartite prior (e.g., a spike at zero speed combined with a Gaussian) to model object motion in the world. The authors should at least discuss this issue, acknowledge the alternatives, and describe why they favor their model.

10) It would be useful to have a clear statement that describes how many free parameters there are in the Bayesian model that is fit to data. One can count them up if they try, but it would be useful to make this easy on the reader. Also, were there any bounds or other constraints on the model parameters in fitting?

Shivkumar S, DeAngelis GC, Haefner RM. Hierarchical motion perception as causal inference.

bioRxiv [Preprint]. 2024 Oct 18:2023.11.18.567582. doi: 10.1101/2023.11.18.567582.

Penaloza B, Shivkumar S, Lengyel G, DeAngelis GC, Haefner RM. Causal inference predicts the transition from integration to segmentation in motion perception. Sci Rep. 2024 Nov 12;14(1):27704. doi: 10.1038/s41598-024-78820-6.

Reviewer #3: In the current study, the authors examined effect of prior visual motion information on vestibular-based self-motion direction judgment. The main finding is that when there is a matching temporal dynamics of visual prior and vestibular signals, the visual would bias the vestibular-based heading judgment. When there is a difference, however, the influence is small. The authors used a Bayesian causal inference model to explain this result. I found this research and finding is very interesting. The conclusion makes sense. The paper is clearly written.

The only unclear part for me is about the three visual conditions with different prior temporal dynamics. Are the three stimulus condition randomly interleaved in one session? It is not clearly described in the method. If they are interleaved, the authors may examine the inter-trial effect, that is, the prior trial on the current judgment.

The other question is that as my understanding, the visual motion is always lateral (left or rightward), while the vestibular motion is mainly upward, which is quite different from the visual in terms of motion direction. So why the subjects cannot exclude the visual stimulus (even identical temporal dynamics), due to a large difference in the direction. If there are trials in which the two modality inputs show similar directions, then this is understandable. Otherwise I may have misunderstood their methods.

The other improvements may be that there could potentially be more operations on the stimulus parameters. For example, reliability of the vestibular cue could be introduced to examine the weight effect. In another example, more stimulus could be introduced between the three visual conditions (e.g. up to 5), to see a more smooth curve of performance under causal inference model. However, I understand these may increase the number of trials, and thus may be considered in future experiments.

**Have the authors made all data and (if applicable) computational code underlying the findings in their manuscript fully available?**

Reviewer #1: Yes

Reviewer #2: Yes

Reviewer #3: Yes

PLOS authors have the option to publish the peer review history of their article (what does this mean? ). If published, this will include your full peer review and any attached files.

**Do you want your identity to be public for this peer review?** For information about this choice, including consent withdrawal, please see our Privacy Policy .

Reviewer #1: No

Reviewer #2: No

Reviewer #3: No

**Figure resubmission:**

**Reproducibility:**



---

## [Decision Letter · Decision Letter 1]

16 Sep 2025

PCOMPBIOL-D-25-00247R1

Environmental motion presented ahead of self-motion modulates heading direction estimation

PLOS Computational Biology

Dear Dr. Kwon,

Thank you for submitting your manuscript to PLOS Computational Biology. After careful consideration, we feel that it has merit but does not fully meet PLOS Computational Biology's publication criteria as it currently stands. Therefore, we invite you to submit a revised version of the manuscript that addresses the points raised during the review process. You should also ensure both data and code underlying the findings described in the manuscript is available in the linked repository.

Please submit your revised manuscript within 30 days Nov 16 2025 11:59PM. If you will need more time than this to complete your revisions, please reply to this message or contact the journal office at ploscompbiol@plos.org. Please include the following items when submitting your revised manuscript:

We look forward to receiving your revised manuscript.

Kind regards,

Paul Bays

Academic Editor

PLOS Computational Biology

Lyle Graham

Section Editor

PLOS Computational Biology

**Reviewers' comments:**

Reviewer's Responses to Questions

Reviewer #1: The authors have thoroughly addressed the reviewer comments and the revised manuscript is greatly improved. I have no further concerns.

Reviewer #2: Overall, the authors have done a good job of responding to the previous reviews and most of my concerns have been addressed adequately. This study has some substantial limitations in addition to multiple strenghts, but I think the limitations are now acknowledged fairly.

One of my major previous concerns was that cue reliability was not varied and so the data may not strongly differentiate a Bayesian model from other types of models that involve a weighted sum of the cues. To address this, the authors have performed formal model comparisons with other models, and Figure 4 is a good addition to the manuscript. That said, I don’t find the results of Fig. 4 to be as definitive as the authors apparently do. Specifically, for the Cov and Fix models, these simpler models perform better than the authors’ CCI model for a substantial minority of subjects. It is true that the average delta_AIC values for these models are positive, but the mean values being positive are pretty strongly driven by a couple of subjects. And for a handful of subjects, the delta_AIC is negative indicating that the simple model is better. I think the authors need to explicitly acknowledge this and soften their conclusions about the superiority of the CCI model.

Partially mitigating the above issue is the new result of Figure 5, which does provide evidence that most subjects are changing their weights. Of course, if the authors had varied cue reliability in some systematic fashion, they could have provided a more powerful test of the superiority of their CCI model. I still see this as a missed opportunity, but I don’t consider it to be a major issue that is fatal here.

Otherwise, the authors have addressed my concerns.

**Have the authors made all data and (if applicable) computational code underlying the findings in their manuscript fully available?**

Reviewer #1: **No: ** I checked the OSF repository and while it contains some data, it does not appear to contain any code to read the data, analyze it, or to reproduce the figures in the paper. The only code is a single short matlab function with the model.

Reviewer #2: None

PLOS authors have the option to publish the peer review history of their article (what does this mean? ). If published, this will include your full peer review and any attached files.

**Do you want your identity to be public for this peer review?** For information about this choice, including consent withdrawal, please see our Privacy Policy .

Reviewer #1: **Yes: ** RM Haefner

Reviewer #2: No

**Figure resubmission:**
---

## [Editor Report · Decision Letter 2]

29 Sep 2025

Dear Dr. Kwon,

We are pleased to inform you that your manuscript 'Environmental motion presented ahead of self-motion modulates heading direction estimation' has been provisionally accepted for publication in PLOS Computational Biology.

Best regards,

Paul Bays

Academic Editor

PLOS Computational Biology

Lyle Graham

Section Editor

PLOS Computational Biology

---

## [Editor Report · Acceptance letter]

PCOMPBIOL-D-25-00247R2

Environmental motion presented ahead of self-motion modulates heading direction estimation

Dear Dr Kwon,

I am pleased to inform you that your manuscript has been formally accepted for publication in PLOS Computational Biology. Your manuscript is now with our production department and you will be notified of the publication date in due course.

With kind regards,

Anita Estes
